# Retinal optic flow during natural locomotion

**Jonathan Samir Matthis**[1]*, **Karl S. Muller**[2], **Kathryn L. Bonnen**[3], **Mary M. Hayhoe**[2]

**1** Department of Biology, Northeastern University, Boston, Massachusetts, United States of America,
**2** Center for Perceptual Systems, University of Texas at Austin, Austin, Texas, United States of America,
**3** School of Optometry, Indiana University Bloomington, Bloomington, Indiana, United States of America

* j.matthis@northeastern.edu

## Abstract

We examine the structure of the visual motion projected on the retina during natural locomotion in real world environments. Bipedal gait generates a complex, rhythmic pattern of head translation and rotation in space, so without gaze stabilization mechanisms such as the vestibular-ocular-reflex (VOR) a walker's visually specified heading would vary dramatically throughout the gait cycle. The act of fixation on stable points in the environment nulls image motion at the fovea, resulting in stable patterns of outflow on the retinae centered on the point of fixation. These outflowing patterns retain a higher order structure that is informative about the stabilized trajectory of the eye through space. We measure this structure by applying the curl and divergence operations on the retinal flow velocity vector fields and found features that may be valuable for the control of locomotion. In particular, the sign and magnitude of foveal curl in retinal flow specifies the body's trajectory relative to the gaze point, while the point of maximum divergence in the retinal flow field specifies the walker's instantaneous overground velocity/momentum vector in retinotopic coordinates. Assuming that walkers can determine the body position relative to gaze direction, these time-varying retinotopic cues for the body's momentum could provide a visual control signal for locomotion over complex terrain. In contrast, the temporal variation of the eye-movement-free, head-centered flow fields is large enough to be problematic for use in steering towards a goal. Consideration of optic flow in the context of real-world locomotion therefore suggests a re-evaluation of the role of optic flow in the control of action during natural behavior.

**Data Availability Statement:** All raw and processed data from this manuscript is available on Figshare at the following DOI https://doi.org/10.25452/figshare.plus.17113883 High resolution. 'mp4' video files for each of the videos in this

## Author summary

We recorded the full body kinematics and binocular gaze of humans walking through real-world natural environment and estimated visual motion (optic flow) using both computational video analysis and geometric simulation. Contrary to the established theories of the role of optic flow in the control of locomotion, we found that eye-movement-free, head-centric optic flow is highly unstable due to the complex phasic trajectory of the head during natural locomotion, rendering it an unlikely candidate for heading perception. In contrast, retina-centered optic flow consisted of a regular pattern of outflowing motion centered on the fovea. Retinal optic flow contained highly consistent patterns that specified the walker's trajectory relative to the point of fixation, which may provide

manuscript are available here - https://doi.org/10.25452/figshare.plus.17121686 The Matlab code necessary to process the data and produce the figures, videos, and simulations used in this study is hosted on GitHub - https://github.com/jonmatthis/RetinalOpticFlow_PLoS_Comp_Bio.

**Funding:** This manuscript was funded by the following sources: NIH NEI 1T32-EY021462 (awarded to MMH), NIH NEI R01-EY05729 (awarded to MMH), and NIH NEI K99/R00-EY028229 (awarded to JSM). The funders had no role in study design, data collection and analysis, decision to publish, or preparation of the manuscript.

**Competing interests:** The authors have declared the no competing interests exist.

powerful, retinotopic cues that may be used for the visual control of locomotion in natural environments. This examination of optic flow in real-world contexts suggest a need to re-evaluate existing theories of the role of optic flow in the visual control of action during natural behavior.

# 1 Introduction

For a mobile creature, accurate vision is predicated on fixation [1]. Vertebrate photoreceptors are relatively slow, with cones taking up to 20 ms to respond to changes in light intensity [2]. Sighted animals must therefore stabilize their eyes relative to the external world to resolve an image that may be used for the control of action. This may be the reason that nearly all vertebrates utilize a "saccade and fixate" strategy whereby gaze is rapidly moved to a new location and then kept stable by way of a gaze stabilization reflexes such as the vestibular ocular reflex (VOR) e.g. [3]. These gaze stabilization mechanisms are phylogenically ancient, with evidence for compensatory eye movements stretching back to the origin of bony fish approximately 450 million years ago [1, 4]. This oculomotor strategy, along with the highly foveated nature of the primate visual system, suggests that the input to the visual system is structured by our ability to rapidly direct gaze to points of interest in the world and then fixate that point as we move our bodies through the world.

This movement of the of the body relative to the point of fixation creates image motion on the retinae, and researchers have long examined the ways that this visual motion (or optic flow) could be used in the control of locomotion. As early as 1950, James Gibson noted that—in the absence of eye movements—an observer moving through through a structured environment will experience a pattern of radial expansion centered on the Focus of Expansion (FoE) that specifies the current heading direction [5, 6]. Subsequent work showed that observers are able to judge their heading in simulated flow fields with very high accuracy, on the order of 1–2 degrees of error [7]. As a consequence of this observation, together with a large body of related work, it became generally accepted that the optic flow patterns arising from linear translation are used by observers to control their direction of heading during locomotion, see reviews by [8–11]. Similarly, this idea has dominated the interpretation of neural activity in motion-sensitive areas of visual cortex such as MST and VIP, areas that respond selectively to optic flow stimuli and are thought to underlie both human and monkey heading judgments [11–18].

However, a complication of this simple strategy is that whenever an individual fixates anywhere but the instantaneous direction of travel, the singularity of the retinal flow field is located at the gaze point rather than the direction of heading (e.g. see Figure 57 in [5]). By definition, the act of fixation nulls image motion at the fovea, so retinal flow during fixation on a stable point in a static scene will always involve patterns of outflowing motion from null motion at the fovea. These fixation-mediated flow patterns were incompatible with the prevailing view that heading perception was derived from eye-movement-free optic flow patterns, so the question of how walkers might recover their heading direction from these retinal optic flowfields became central to much of the research on optic flow (dubbed the "rotation problem" (e.g. Britten 2008 [11]) or even the "eye movement problem" [19]). Solutions tended to involve decomposing of the retinal flow patterns into a rotational component arising from eye movements and a translational component arising from locomotion (typically simulated as straight line constant velocity trajectories). In the large body of subsequent research, it was shown that subjects can make accurate judgements heading direction during eccentric fixation

(e.g. [20–25]), which was taken as support for this interpretation of the role of optic flow in the control of locomotion.

However, regardless of what the eyes do during locomotion, the complex, phasic movements of the head throughout the bipedal gait cycle will strongly increase the variability in the optically specified instantaneous heading direction. The stimuli typically used in studies of optic flow simulate constant velocity straight-line movement. These stimuli produce a strong sense of self-motion, or vection, and are a reasonable approximation to the airplanes, automobiles, and gliding birds that were the basis of Gibson's original insights. However, natural locomotion results in rhythmic translation and rotation profiles of the head in space, peaking at approximately 2Hz, which is the natural period of footsteps [26, 27]. This means that the momentary heading direction varies through the gait cycle, creating a complex pattern of flow on the retina. While this has long been recognized (e.g. [28–30]), the gait induced oscillations are typically assumed to be a minor factor and are referred to as "jitter" (e.g. [31–33]). During natural locomotion, gaze patterns vary depending on the complexity of the terrain [34, 35], and the gait-induced trajectory of the head and the terrain-dependent patterns of eye movements during natural locomotion determine the actual flow patterns on the retina. There has been some exploration of retinal motion patterns by measuring eye and head movements during locomotion [30, 35, 36]. However, those measurements focused on the statistics of retinal flow, rather than the time-varying evolution of the signals. Many algorithms exist to recover observer translation and rotation from the instantaneous retinal flow field. However, there is disagreement on whether heading perception is based on the instantaneous flow field, or instead based on some measure of the way the flow patterns change over time [37–40].

The goal of this paper is to measure eye, body, and head movements during natural locomotion and to use this data to investigate the resulting optic flow patterns. We first calculated the flow patterns relative to the head, as this reflects the way that the movement of the body during gait impacts instantaneous heading direction by showing an eye-movement-free representation of optic flow. Then, we combine these head-centered flowfields with measured eye position to estimate the retinal optic flow experienced during natural locomotion. By characterizing the optic flow stimulus experienced during natural locomotion, we may gain a greater insight into the ways that the nervous system could exploit these signals for locomotor control.

We recorded subjects walking in natural environments while wearing a full-body, Inertial Measuremen Unit (IMU)-based motion capture suit (Motion Shadow, Seattle, WA, USA) and a binocular eye tracker (Pupil Labs, Berlin, Germany). Data from the two systems were synchronized and spatially calibrated using methods similar to those described in [34] (See Methods and materials for details). This generated a data set consisting of spatiotemporally aligned full-body kinematic and binocular gaze data (sampled at 120Hz per eye), as well as synchronized first-person video from the head-mounted camera of the eye tracker (100 deg. diagonal, 1080p resolution, 30Hz). The integrated data are shown by the skeleton and gaze vectors in Fig 1B and S1 Video (A tabulated list of all videos in this manuscript may be found in Table 1). Subjects walked in two different environments: **Woodchips**—a flat tree-lined trail consisting of mulched woodchips shown in Fig 1 (selected because it was highly visually textured, but flat enough that foot placement did not require visual guidance); and **Rocks**—a rocky, dried creek bed, where successful locomotion requires precise foot placement (This was the same terrain used in the 'Rough' condition of [34]).

Subjects walked the woodchips path while completing one of three experimental tasks:

**Free Viewing**—where subjects were asked to walk while observing their environment with no explicit instruction;

**Ground Looking**—where subjects were asked to look at the ground at a self-selected distance ahead of them (intended to mimic gaze behavior on the rocky terrain without the

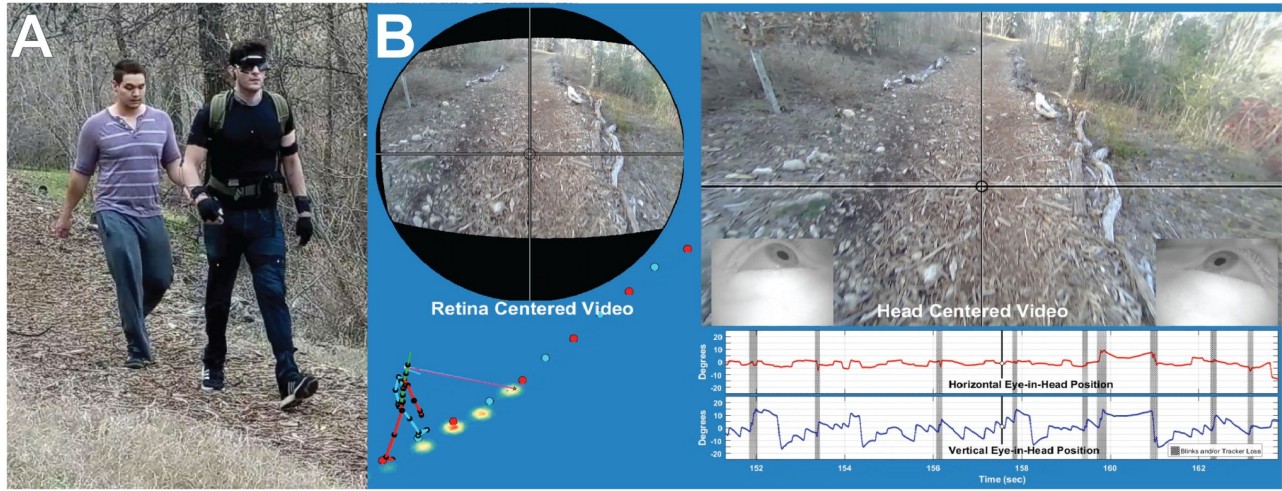

**Fig 1. The data collection setup.** (A) shows the subject walking in the Woodchips terrain wearing the Pupil Labs binocular eye tracker and Motion Shadow motion capture system. Optometrist roll-up sunglasses were used to shade the eyes to improve eye tracker performance. (B) shows a sample of the data record, presented as a sample frame for S1 Video. On the right is the view of the scene from the head camera, with gaze location indicated by the crosshair. Below that are the horizontal and vertical eye-in-head records, with blinks/tracker losses denoted by vertical gray bars. The high velocity regions (steep upwards slope) show the saccades to the next fixation point, and the lower velocity segments (shallow downwards slope) show the component that stabilizes gaze on a particular location in the scene as the subject moves towards it, resulting a characteristic saw-tooth appearance for the eye signal (without self-motion and the associated stabilizing mechanisms these saccades would exhibit a more square-wave like structure). On the left, the stick figure shows the skeleton figure reconstructed form the Motion Shadow data. This is integrated with the eye signal which is shown by the blue and pink lines. The representation of binocular gaze here shows the gaze vector from each eye converging on a single point (the mean of the two eyes). The original ground intersection of the right and left eye is shown as a magenta or cyan dot (respectively, more easily visible in S1 Video). The blue and red dots show the foot plants recorded by the motion capture system. The top left figure shows the scene image centered on the point of gaze reconstructed from the head camera as described in the Methods and Materials section.

**Table 1. List of videos in this manuscript.**

| Video ID | Eye Trajectory / Video source | Data type | Short Description |
|---|---|---|---|
| S1 Video | Real-world gaze/kinematics | Raw Laser Skeleton | Woodchips—Free Walking (Full speed, See Fig 1) |
| S2 Video | Real-world gaze/kinematics | Optic Flow—Vectors | Woodchips—Free Walking (Quarter Speed) |
| S3 Video | Real-world gaze/kinematics | Optic Flow—Streamlines | Woodchips—Free Walking (Quarter Speed) |
| S4 Video | Simulated Eye Trajectory | Simulated Retinal Flow | Fixation Aligned with Path |
| S5 Video | Simulated Eye Trajectory | Simulated Retinal Flow | Fixation to Left of Path |
| S6 Video | Simulated Eye Trajectory | Simulated Retinal Flow | Fixation to Right of Path |
| S7 Video | Simulated Eye Trajectory | Simulated Retinal Flow | Vertical Sin Wave |
| S8 Video | Simulated Eye Trajectory | Simulated Retinal Flow | Horizontal Sin Wave |
| S9 Video | Simulated Eye Trajectory | Simulated Retinal Flow | Forward Corkscrew |
| S10 Video | Real-world gaze/kinematics | Simulated Retinal Flow | Woodchips—Ground Looking (Quarter speed) |
| S11 Video | Real-world gaze/kinematics | Raw Laser Skeleton | Rocky Terrain (Full speed) |
| S12 Video | Real-world gaze/kinematics | Optic Flow—Streamlines | Rocky Terrain |
| S13 Video | Real-world gaze/kinematics | Simulated Retinal Flow | Rocky Terrain |
| S14 Video | Quadcopter Gimbal Video | Optic Flow—Streamlines | Drone video validation |

Each Video ID is a clickable link to a YouTube video. A YouTube playlist of all videos in the manuscript is here—https://www.youtube.com/playlist?list=PLWxH2Ov17q5HRHVngfuMgMZn8qfOivMaf. These videos exist in a long-term stable archive on Figshare—doi://10.25452/figshare.plus.17121686. Extended captions for each video is available in the Supporting Information section.

complex structure or foothold constraints). Note that although previous research has suggested that walkers may engage in "travel gaze" during locomotion (where gaze is "visually anchored in the front of the individual and carried along by the whole body movement [41]"), this behavior is inconsistent with gaze stabilization reflexes, such as the VOR and associated spinal modulation, OKN, and potentially predictive pursuit (e.g. [42–44]). Consequently, in this task subjects' gaze patterns always consisted of a series of brief fixations and small saccades to keep gaze roughly a fixed distance ahead of the walker.

**Distant Fixation** wherein subjects were asked to maintain fixation on a self-selected distant target that was roughly at eye height (this condition was intended to most closely mimic psychophysical tasks that have often been employed to explore perception of heading, [8, 19, 45, 46]). Note we use the term "fixation" here to refer to the periods of when gaze is stably fixed to a point in the world, even when there is some rotation in the orbit as a consequence of stabilization mechanisms. Although fixation is often used to refer to periods when the eye is fully stable within its orbit, these moments are extremely rare during natural behavior due to the constant movement of the head (and the resulting reflexive gaze stablizing eye movements).

Walking over the rocky terrain (**Rocks**) was demanding enough that subjects were not given instructions other than to walk from the start to the end position at a comfortable pace. Note that these conditions were primarily selected because the behaviors they represent provide opportunities for later analysis. In most cases, data gathered in the various conditions were not notably different in the dimensions explored in this manuscript (e.g. Fig 2). Detailed analysis of behavioral differences between conditions is beyond the scope of this paper.

## 2 Optic flow during real-world locomotion

### 2.1 Effect of gait on head-centered optic flow

To measure the optic flow patterns induced by the body motion independently of eye position, we first ran the videos from the head-mounted camera of the eye trackers through a computational optic flow estimation algorithm DeepFlow [47], which provides an estimate of image motion for every pixel of each frame of the video (S2 Video). As an index of heading we tracked the focus of expansion (FoE) within the resulting flow fields using a novel method inspired by computational fluid dynamics (See Methods and materials). This analysis provides an estimate of the FoE location in head-centered coordinates for each video frame (S3 Video).

We found that the head-centered optic flow pattern is highly unstable, rarely corresponds to the walker's direction of travel, and never lies in a stable location for more than a few frames at a time Fig 3. The FoE moves rapidly across the camera's field of view at very high velocities with a modal velocity across conditions of about 255 deg per sec (Fig 2). Note that this analysis is entirely based on recorded video from the head-mounted camera and does not rely on the IMU measurements. To ensure that this instability is not just a consequence of the video-based technique, we also measured the FoE in the video of a gimbal-stablized quadcopter camera drone (DJI Phantom 4, Nanshan, Shenzhen, China) and found it to be stable (See Methods and materials). Thus the motion of the FoE is not an artifact of the method used here.

The instability of the head-centered flow can be seen in Fig 3, which shows the head-centered optic flow over a period of 150ms (5 frames at 30fps). The red line traces the movement of the FoE over the 5 frames (150ms). Fig 2 in the figure shows the velocity distribution for the FoE, where the average of the 4 conditions is shown in black, and the distributions for the different conditions is shown by the colored curves. Note that the instability of the FoE is relatively unaffected by either the terrain (rocks or woodchips) or by subject's instructions for gaze behavior (distant fixation vs ground looking). This may be because the basic movement patterns of bipedal locomotion were not altered across the different conditions, so the resulting

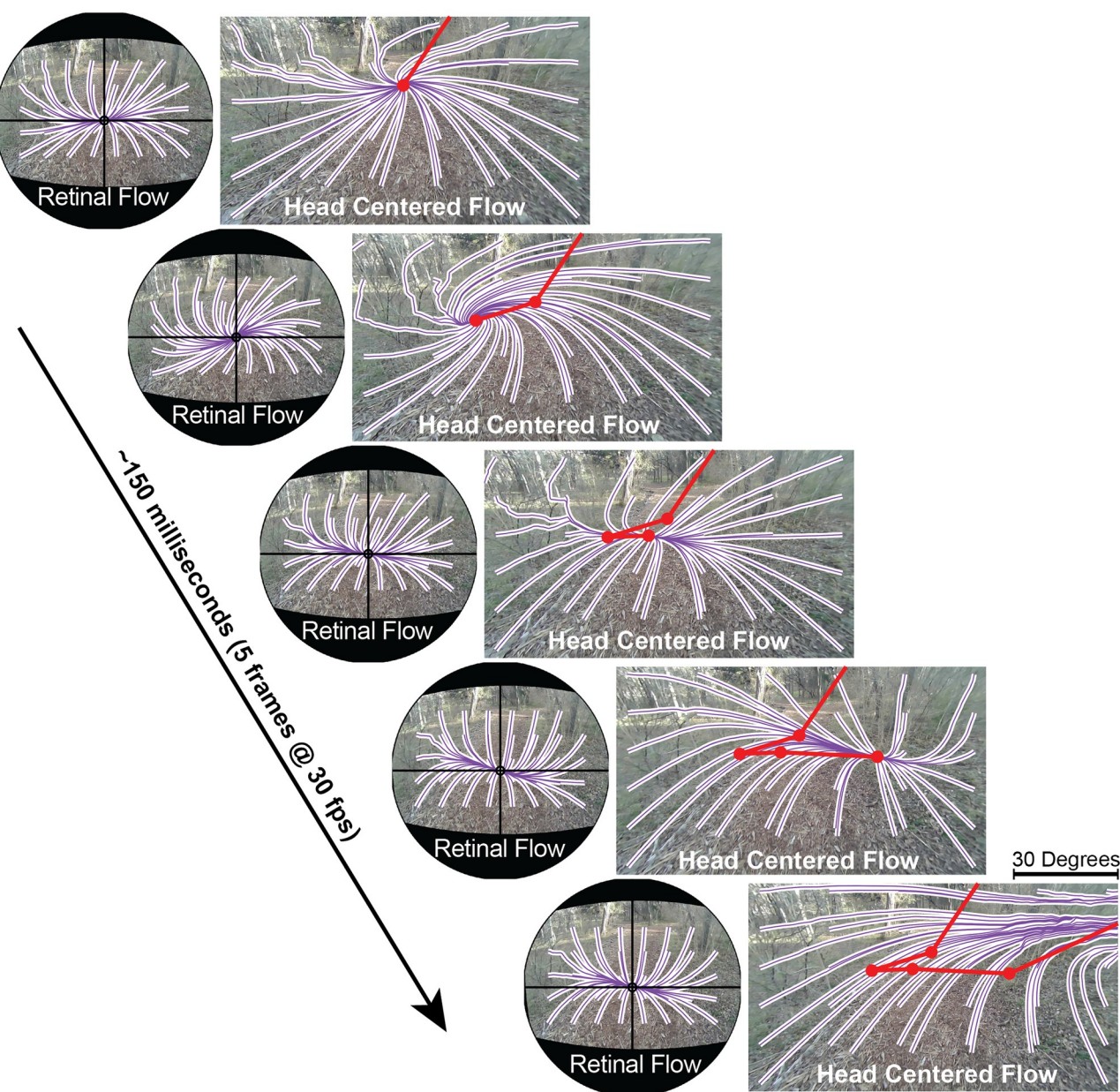

**Fig 2. Retinal vs head-centered optic flow.** Optic flow patterns (down-sampled) for a sequence of 5 video frames from S3 Video, altered for visibility in print. Head centered flow shows optic flow in the reference frame of the head mounted ";world" camera, and represents optic flow free from the effects of eye movements. Retinal flow shows optic flow in the references frame of a spherical pinhole camera stabilized on the subject's fixation point. Purple and white lines show the integral curves of the measured flow fields, determined by using the streamlines2 function in Matlab to make a grid of particles drift along the negation of the flow fields measured by the DeepFlow optical flow detection algorithm in OpenCV. The red trace shows the movement of the head-centered FoE moving down and to the right across the 5 frames. In contrast, note the stability of the retina-centered flow over the same period.

statistics of FoE velocity would be similarly conserved. The instability of the head-centered FoE is also clearly demonstrated in S3 Video, which shows the integral lines of the optic flow vector fields shown in S2 Video with the FoE on each frame denoted by a yellow star. This lack of stability means that recovering heading during locomotion cannot be achieved by simply accounting for eye position. Head trajectory during locomotion cannot be approximated by

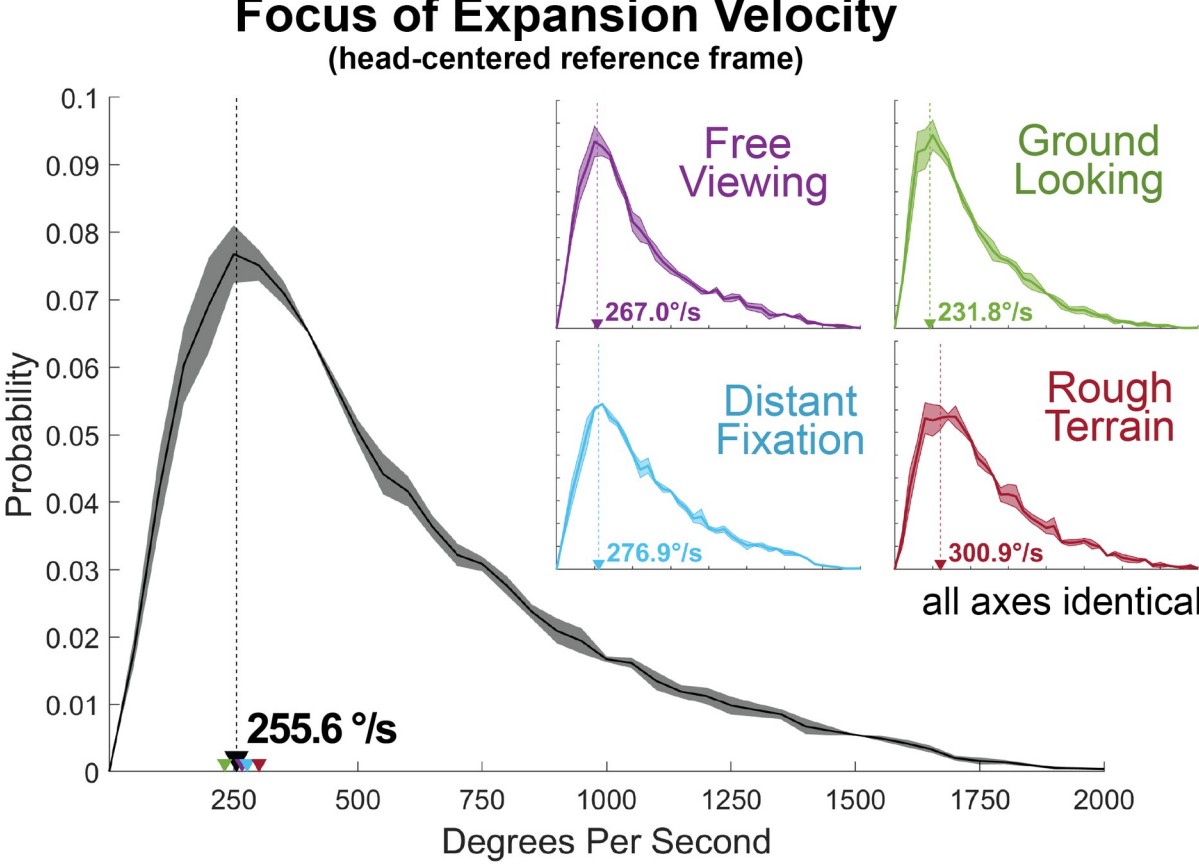

**Fig 3. Focus of Expansion velocity in head-centered coordinates.** Focus of Expansion velocity across all conditions (black histogram), as well as split by condition (colored insets). The thick line shows the mean across subjects, and shaded regions show +/-1 standard error.

constant velocity, straight line motion (e.g. [8, 19, 45, 46]) or with added 'jitter' [33]. To better understand the reason for the extremely high velocity of the FoE in a head-centered reference frame, we must examine the trajectory of the head throughout the gait cycle (Fig 4).

## Head velocity oscillations during natural locomotion

The instability of head-centered optic flow arises from the natural oscillations of the head during locomotion. A walker's head undergoes a complex, repeating pattern of velocity throughout the gait cycle, as shown in Fig 4. Although the vestibulocollic and spinocollic reflexes result in some damping of head acceleration in the Anterior/Posterior and Medial/Lateral directions (note the flattening of the acceleration profile from hips to chest, and chest to head in these dimensions), no such damping appears to occur in the vertical direction, most likely because it would interfere with the efficient exchange of potential and kinetic energy inherent in the inverted pendulum dynamics of the bipedal gait cycle [48–51]. As a result, a walker's head undergoes a continuously changing phasic velocity profile over the course of each step (see yellow vector and trace in S3 Video, which shows the velocity vector of the walker's eyeball center at each frame of the recording). The peak magnitude of these velocity oscillations (particularly in the vertical dimension) is approximately half the walker's overall walking speed and so will have a substantial effect on the stability of head-centered optic flow. These variations in head velocity (Fig 4) and the accompanying rotations (S1 Fig) throughout the gait cycle are the

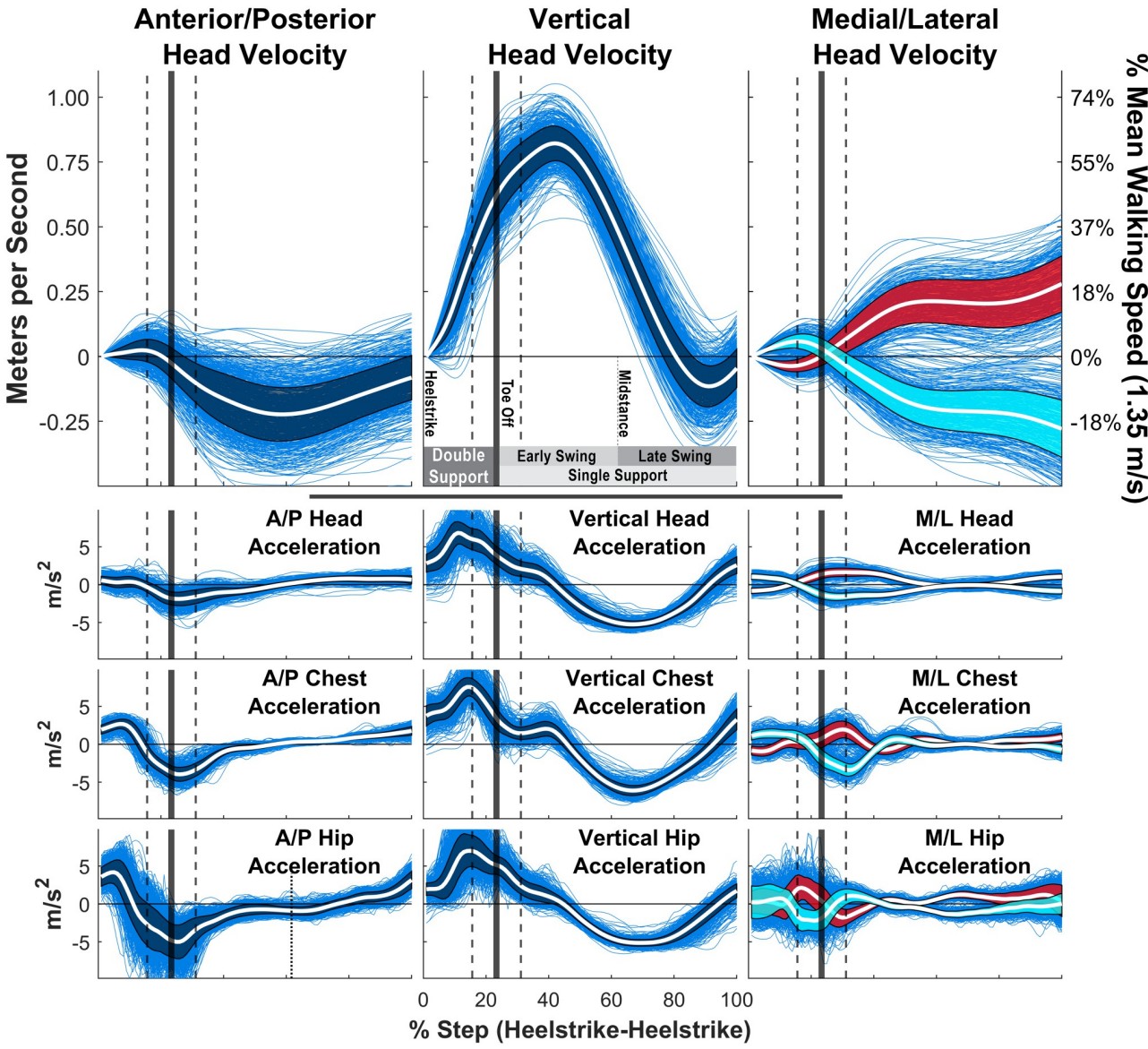

**Fig 4. Head velocity and acceleration during locomotion.** Head velocity and head/chest/hips acceleration patterns of a single representative subject walking in the "Distant Fixation" condition (where they walk while maintaining fixation on a point far down their path). Each blue trace is the velocity or acceleration trace for a single step, normalized for the time period from heel-strike to the subsequent heel-strike. Acceleration traces derive from the triaxal accelerometers of the subjects' IMU sensors, while velocity traces are derived from numerical integration of these signals. The white line shows the mean and the shaded region is +/- 1 standard deviation. Right and Left steps were computed separately for the Medial/Lateral data, and are shown in red and cyan respectively. The vertical solid lines show mean toe-off time, with the vertical dashed lines showing +/- 1 standard deviation.

reason why the head-centered flow patterns move at such high velocities across a walker's visual field.

The location of the FoE at any given moment depends on the head's velocity vector [20], so changes in that vector will lead to changes in the FoE's location in the visual field. Furthermore, because the FoE arises from the visual motion of objects in the walker's field of view, the linear velocity of the FoE in the viewing plane of the walker will be determined by the angular variations in the walker's head velocity vector projected onto the visible objects within walker's environment. Consequently, small angular changes in the walker's head

velocity result in rapid, large scale movements in the FoE on the walker's image plane in a way that depends of the depth of the scene in the direction of the head's translation vector. This insight in conjunction with observation of the rapid changes in the head velocity vector throughout the gait cycle (the yellow vector in S3 Video) are the reason for the extremely high FoE velocities observed in the present data (Fig 4, and the chaotic FoE paths seen in S3 Video and Fig 3). While the importance of the FoE itself has been questioned [8, 19], the instability applies to the entire flow field. Any theory for heading perception during locomotion the relies on eye-movement free representations of optic flow must be robust to this high degree of FoE instability.

## Retinal optic flow during natural locomotion

To estimate retinal flow patterns during natural locomotion, we first identified fixations in the eye tracker data by locating the beginning and end of the saccades using an acceleration criterion. During the interval between saccades the eye is approximately stable at a location in the environment, but may still move in the orbit as to account for the movement of the body. Analysis of the scene camera images revealed excellent stabilization, which is elaborated in the Methods and Materials For computational simplicity, we assume perfect stabilization during a fixation, and aligned each frame from the head-mounted video so that the point of fixation was always pinned to the center of the image (see Methods and materials). Then, the resulting 'gaze stabilized' videos were analyzed with the same optic flow and streamline based analysis used for the head-centered videos. The Methods and Materials includes an analysis the extent of imperfect stabilization and calculate the consequences for our estimates of retinal flow. The conclusions we make do not depend on getting exact estimates of retinal flow.

A comparison of retinal and head centered flow patterns during a single fixation is shown in Fig 3. It can be seen that the retinal patterns change only modestly over time as a consequence of the gaze stabilization. This suggests that gaze stabilization/fixation may be an important component in understanding the use of optic flow patterns, as has been suggested by [30, 52, 53] rather than a complicating factor as has often been assumed (e.g. [11]). When gaze is stabilized on a static location in the world during locomotion, the result will always be a pattern of radial expansion centered on the point of fixation, with particular features (e.g. spiral patterns and parallax deformation) arising from complexities in the observer's movement or the structure of the scene (e.g. the Retinal Reference Frame panel of S3, S9 and S12 Videos). This suggests that valuable information for the control of locomotion might exist in the higher order structure of these fixation-mediated flow patterns. In what follows, we explore some of the task relevant features of retinal flow and suggest how they might be used for the control of locomotion.

## 3 Simulating retinal flow from kinematic data

The preceding analysis used computational analysis of videos from a mobile eye tracker estimate head-centered and retinal optic flow during natural locomotion. In this section, we perform a complimentary investigation of the structure of the retinal motion patterns using the walkers' recorded gaze and kinematic data to geometrically simulate the flow patterns experienced during natural locomotion.

These simulations of retinal flow utilize **a spherical pinhole camera model of the eye** (Fig 5). The spherical eye maintains 'fixation' on a point on the ground, while a grid of other ground points is projected through the pupil and onto the back of the spherical eye. The locations of these projected ground points on the retina are tracked frame-by-frame to calculate

## Spherical Pinhole Camera Eye Model

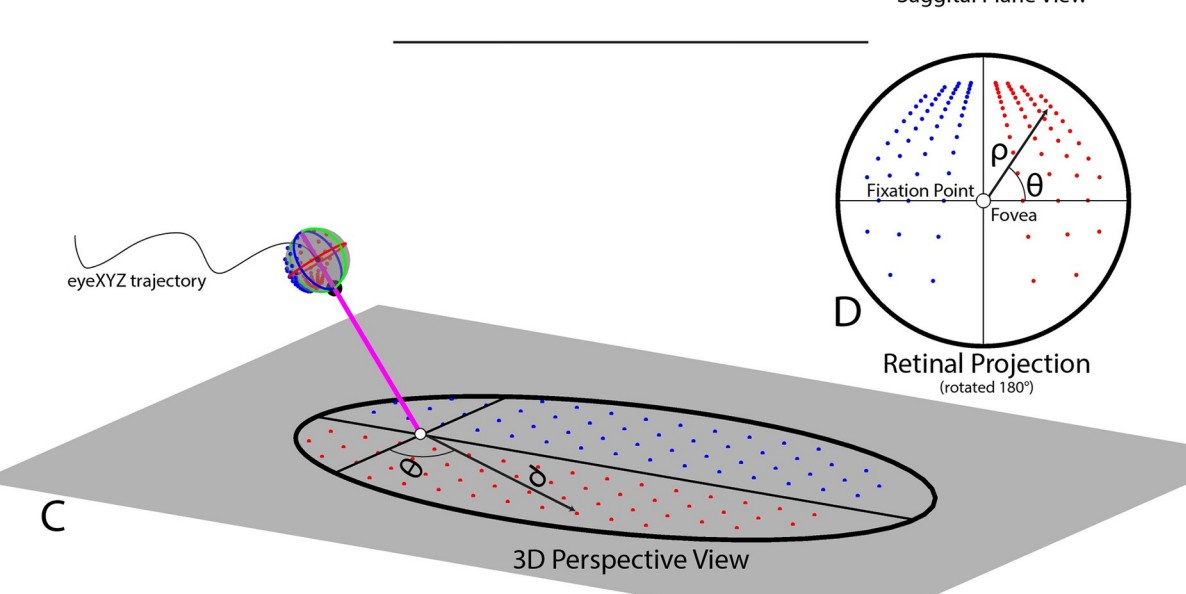

**Fig 5. Spherical pinhole camera model of the eye.** The spherical pinhole camera model of the eye used to estimate retinal optic flow experienced during natural locomotion. A, B show a sagittal plane slice of the 3D eye model. A shows the eye fixating on a point on the ground (pink line shows gaze vector, black circle shows fixation point) as points in the upper (orange) and lower (green) visual fields project on the back of the eye after passing through a pinhole pupil. B shows a closer view of the sagittal slice of the eye model. C, D show the full 3D spherical pinhole eye model. C shows the 3D eye fixating a point on the ground (black crosshairs), with field of view (60 degree radius) represented by the black outline. Note that the circular field of view of the eye is elongated due to its projection onto the ground plane. Red and blue dots represent points in the right and left visual field, respectively. D shows the retinal projection of the ground points from C on the spherical eye model. Ground dot location in retinal projection is defined in polar coordinates (ϑ, ρ) relative to the fovea at (0,0), with ϑ defined by the angle between that dot's position on the spherical eye and the 'equator' on the transverse plane of the eye (blue circle) and ρ defined as the great-circle (orthodromic) distance between that dot and the fovea of the spherical eye. The retinal projection has been rotated by 180 degrees so that the upper visual field is at the top of the image.

simulated retinal optic flow. The eyeball maintains fixation on a defined location on the groundplane while following an arbitrary trajectory through space to determine the structure of retinal optic flow during various types of movement. In S4–S9 Videos the eye model was set to follow pre-specified trajectories while maintaining fixation on a single location to explore and highlight different aspects of retinal flow. In S10 and S13 Videos, this model of the eye was set to follow the 3-dimensional trajectory of walker's eyes recorded from the motion capture suit as they walked in natural terrain. These simulations allow us to estimate retinal motion experienced during natural locomotion without the complicating elements associated with the computational analysis of real-world video recordings (see Methods and materials for details).

### 3.1 Retinal cues for the control of locomotion

Stabilization of gaze during fixation nulls visual motion at the fovea, so the basic structure of retinal optic flow will always consist of outflowing motion centered on the point of fixation. The retinal motion results from the translation and rotation of the eye in space, carried by the body and the walker holds gaze on a point on the ground during forward movement. We found several features of the retinal flow patterns that provide powerful cues for the visual control of locomotion, which we describe below.

### 3.2 Consequences of residual image motion from imperfect fixation

If gaze is not perfectly stationary during fixation, there will be a small translational component added to the retinal motion. Unfortunately, a full explication of this effect is beyond the scope of this paper, as the error associated with incomplete gaze stabilization during fixation is likely a similar magnitude to the errors expected from the contemporary of mobile eye trackers. However, we estimated the magnitude of this slip and found that stabilization during a fixation is quite good, with the median slip being less than a degree of visual angle along the direction of travel during a fixation, with the gain being slightly less than 1.0. For this reason, we chose to "idealize" fixations by nulling motion at the fovea to pin the gaze vector to the groundplane during each fixation and kept our research questions to a scale that will not be sensitive to this simplification (see Methods and materials for more).

### 3.3 Foveal curl provides a cue for locomotor heading relative to the fixation point

When a walker fixates a point that is aligned with their current velocity vector, the resulting retinal flow field has no net rotation at the fovea (Fig 6A, S4 Video). However, fixation of a point that is off of their current trajectory results in a rotating pattern of flow around the fovea that may be quantified by calculating the curl of the retinal vector field. Fixation of a point that is to the left of the current path results in counter-clockwise rotation (Fig 6B, S5 Video), while fixation to the right of the current trajectory results in clockwise rotation around the fovea (Fig 6C, S6 Video). This shows that the curl of retinal flow around the fovea provides a metric for the walker's trajectory relative to their current point of fixation.

Thus, the magnitude of rotation around the point of fixation provides a quantitative cue for the walker's current trajectory relative to the point of fixation. Specifically, there will be zero net curl around the fovea when the walker is fixating a point that is aligned with their current trajectory, and positive/negative curl values for fixation to the left/right of the current trajectory, with the magnitude of the curl signal increasing proportional to the eccentricity of the fixation point (S8 Video). This cue could be used to help the walker move the body in a particular direction relative to the current fixation location on the ground. Maintaining fixation on a specific location and turning until there is zero curl around the fovea would be an effective way for a walker to orient the body towards a specific location (such as a desirable foothold in the upcoming terrain), correct for potential postural errors, or simply to maintain a straight heading by balancing the horizontal trajectory changes that occur during steps on the right or left foot (S10 Video). Similarly, walkers might learn to predict a certain pattern of retinal flow for a given bodily movement, and values outside this range could provide a signal for balance control [54–56]. While the curl component of retinal flow has been well established [8, 19, 31, 57–59], those conversations typically focus on the use of optic flow in longer term heading perception (e.g. steering the body towards a distant target). The present examination of this time varying signal for the control of the shorter-term, biomechanical aspects of locomotion (e.g.

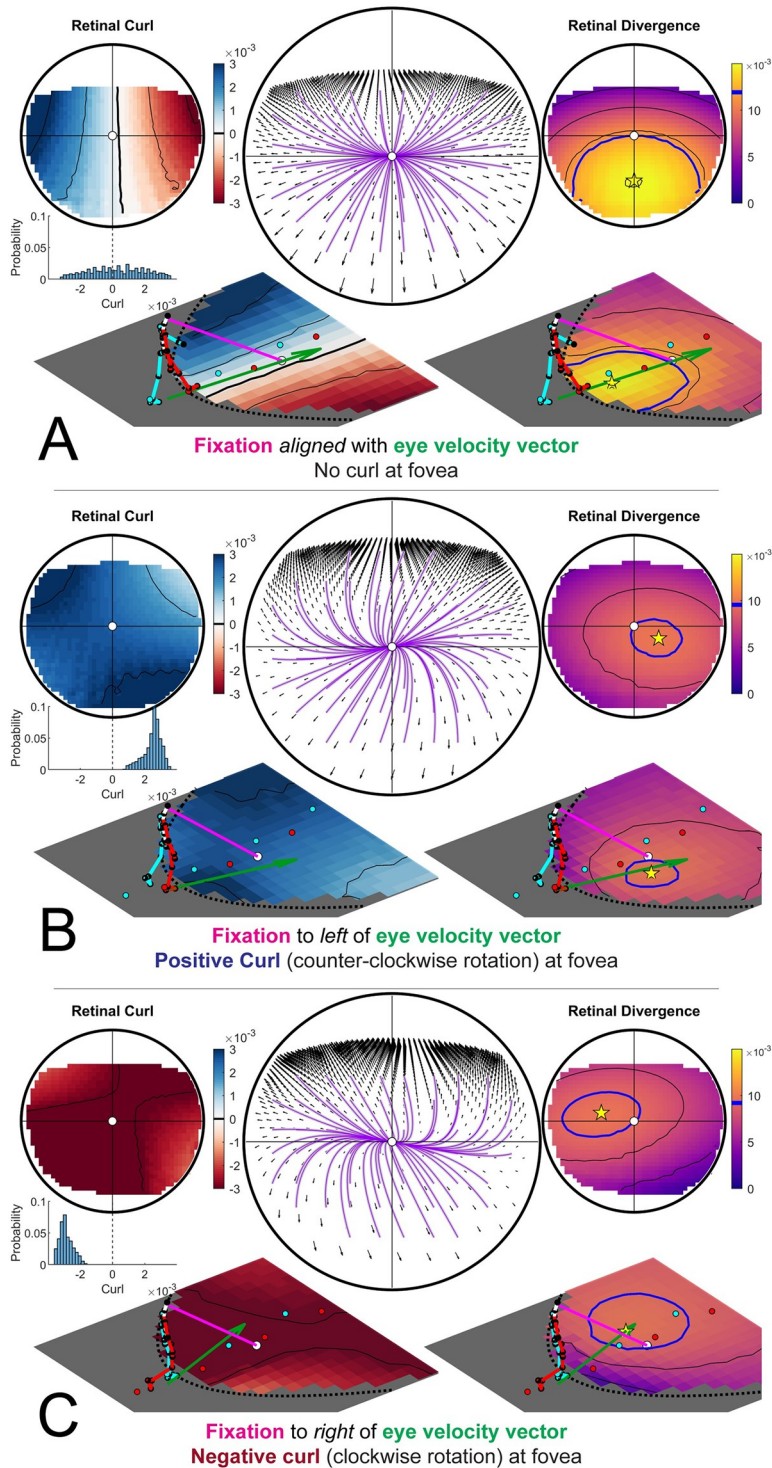

**Fig 6. Retinal optic flow during natural locomotion.** Retinal flow simulation based on integrated eye and full-body motion of an individual walking over real world terrain. Panels A-C are based on frames taken from S2 Video, which shows data derived from a subject walking in the Woodchips condition under the Ground Looking instructions (S10 Video). (A) shows a case where the fixation point (pink line) is aligned with ground projection of the eye's velocity vector (green arrow). Middle circular panel shows simulated optic flow based on fixation location and body movement. Left and right upper circular panels show the results of applying the curl and divergence operators to the retinal flow field in the middle panel. Left and right bottom panels show the projection of the curl (left) and divergence (right) onto a flat ground plane. The green arrow shows the walker's instantaneous velocity vector (scaled for

visibility), which always passes through the maximum of the retinal divergence field(which always lies within the foveal isoline (blue circle)) (see Fig 7). (B) and (C) show cases where the fixation point is to the left or right of the eye's velocity vector (respectively). Fixation to the left of the eye's velocity vector (B) results in a global counter-clockwise rotation of the retinal flow field and positive retinal curl at the fovea, while fixation to the right of the eye's velocity vector results in clockwise flow and negative curl at the fovea (Fig 7).

foot placement and postural maintenance in rocky terrain) strongly suggests a different role for the signal.

### 3.4 The point of maximum divergence encodes the body's over-ground momentum vector in retinotopic coordinates

Fixating a point on a point on the ground while moving forward creates a pattern of expansion centered around the point of fixation. The local expansion at each point of a vector field can be quantified by calculating the *divergence* of the vector field. Divergence can be intuitively thought of as the rate that an array of drifitng particles would vacate each point in the flow field. In contrast to optic flow on a fronto-parallel plane, the distance gradient resulting from fixation on the ground plane results in variation in motion parallax of points on the ground plane across the upper to lower visual field. Therefore, divergence of the resulting flow field is determined by the velocity of the eye relative to the fixated point combined with the parallax gradient of points on the ground plane [58]. In the context of locomotion, the divergence field during a ground plane fixation results in a hill-like gradient with a peak denoting the point of maximum retinal divergence that is contained within the ellipsoidal isoline that passes through the point of fixation (green circle in Fig 6). During straight-line movement towards the point of fixation the foveal iso-ellipsoid begins in the lower visual field before constricting to a point when the observer is one eye height away from their point of fixation (that is, when their gaze angle is 45 degrees) and then expanding into the upper visual field (Fig 6A–6C, S4–S6 Videos, Note that this feature of the divergence field is affected by upward and downward trajectories of the eye (S7 Video).

Projecting the retinal divergence map onto the ground plane reveals that the ground projection of the eye's velocity vector always passes through the point of maximum retinal divergence (Fig 7B). This feature of retinal flow means that a walker could, in principle, discern their own world-centered ground velocity in retinotopic coordinates directly from retinal flow. Thus it should be possible to determine the body's world-centered velocity vector directly from patterns of motion on the retina. Because the walker's mass is constant, this velocity vector may serve as a proxy for the body's momentum vector in retinotopic coordinates. The body's momentum vector is the information a walker needs to make biomechanically informed decisions about where to place their feet so the ability to derive this information directly from retinal optic flow may be of critical importance for the control of locomotion. Since the walker can determine the body-relative location of their fixation point [60], this analysis shows how patterns of stimulation on the retina might be directly related to the physical dynamics of bipedal locomotion [48, 61–64].

The correspondence of the projection of the eye's velocity vector onto the ground plane with the point of maximum divergence was previously described by Koenderink and van Doorn, [58]. The use of this information as a cue for heading was subsequently challenged by Warren and Hannon [19]. In one of the five behavioral experiments described in that paper, Warren and Hannon asked stationary subjects viewing a computer monitor to judge their heading during simulated motion through a 3D star field. The authors argue that because these star fields were spatially discontinuous, the resulting flow fields will not contain a point

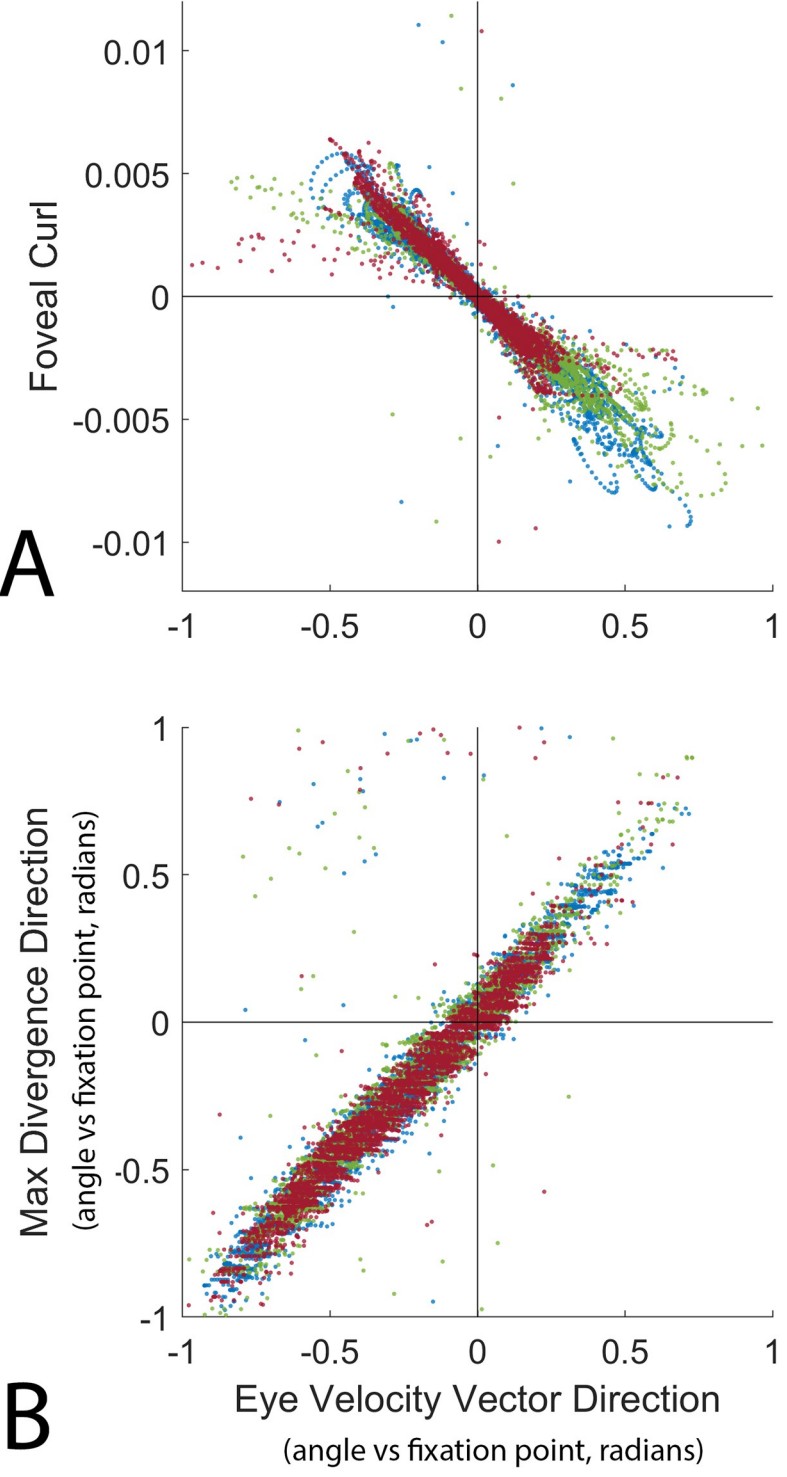

**Fig 7. The relationship between curl, divergence, and the angle between the eye's velocity vector and the fixation point.** The x-axis in A & B shows the angle between the vertical projection of the eye's velocity vector and the fixation point, with zero denoting that the eye's instantaneous velocity vector passes directly through the point of fixation on the ground, negative values mean the eye will pass to the left of the fixation point, and positive values mean the eye will pass to the right. The Y-axis of (A) represents the curl of the retinal flow field at the fovea (foveal curl). Each dot represents a single recorded frame from S10 Video, with the different colors denoting data from the three subjects. The Y-axis of (B) shows the angle between the point of maximum divergence in the retinal flow field and the point of fixation, measured similarly to the angle between the eye vector and the fixation point on the x-axis.

of maximum divergence. They took the finding that subjects made accurate heading estimates in these star fields as evidence that humans do not use the point of maximum divergence to determine their heading direction.

There are at least three problems with this line of argument. First and foremost, the finding that subjects can make accurate heading judgements in the absence of a particular cue is not evidence that they do not use that cue when it *is* present. Secondly, even if it is the case the the stimuli presented to subjects were sparse or discontinuous to the point where divergence was undefined, this does not mean that the stimuli encoded by subjects would have the same properties. The observer judgments have been shown to be robust to sparse, discontinuous motion stimuli (e.g. [65, 66]), so the mathematical properties of the experimental stimuli may not match the stimuli that make it through the complex neurophysiological filters of the visual system. Finally, and relating to the first point, the stimuli used in this study were decidedly unrealistic and do not represent a natural occurring pattern of visual motion, so this study should not be taken as strong evidence against the use of retinal divergence for the control of action. However, it is worth reiterating that the present, purely observational investigation also does not provide any evidence that walkers use divergence in the control of locomotion. Rather, we simply show that the the point of maximum divergence is a reliable feature of fixation-mediated retinal optic flow that appears to encode behaviorally relevant information that *could* be valuable for the control of locomotion. The question of whether walkers actually use this cue— or even if the visual system is physiologically capable of detecting it—must be a matter of experimental investigation.

In addition, it should be noted that this simulation assumes a flat ground plane. In rocky or irregular terrain, local structure from motion will add complex local variations to the underlying retinal motion patterns described here, and it remains to be explored how these complexities might influence locomotion. However, walkers have ample opportunity to learn the statistical patterns associated with locomotion over a variety of terrains, as well as the range of patterns associated with stable postures. It is likely that retinal motion cues from both eyes are combined to provide a robust estimate of both our movement through the 3D structure of the world, in what has been referred to as the "binoptic flow field" [67, 68]. Nonetheless, the monocular task-relevant features of retinal optic flow described here is robust to complex, sinusoidal and circular simulated eye trajectories (S9 Video), locomotion over flat terrain (S10 Video), as well as the more circuitous routes taken in the rocky terrain (S12–S14 Videos).

## 4 Discussion

We have measured eye and body movements during locomotion in natural terrain and from this, described the ways that the optic flow patterns are shaped by the movement of the observer through the environment. The natural variation in heading during the gait cycle means that the flow pattern relative to the head is highly unstable. This is true whether the walker's head (and eye) is directed towards a distant target or at the ground nearby to monitor foothold selection. The problem of heading perception is typically characterized as one of accounting for eye movements to recover the translational direction of the head. Some analyses posit an explicit representation of motion signals relative to the head (see [25, 31, 46]). In this case, the variation in the motion patterns relative to the head during the gait cycle make it unlikely that these signals could be used to control steering towards a goal. However, robust stabilization of gaze means that retinal motion patterns during the gait cycle are much less variable and exhibit regularities that may be valuable for the control of locomotion. In particular, a walker can determine whether they will pass to the left or right of their fixation point by

observing the sign and magnitude of the curl of the flow field at the fovea. In addition, the divergence map of the retinal flow field provides a cue for the walker's over-ground velocity/ momentum vector, which may be an essential part of the visual identification of footholds during locomotion over complex terrain. The magnitude of the image velocities relative to the head make it unlikely that instantaneous heading is useful for control of navigation involving steering towards a distant goal.

The geometric relationship between retinal optic flow patterns and the translation of the eye over a groundplane was carefully investigated in a series of papers by Koenderink and van Doorn (e.g. [58, 59, 69]). This work also suggested that retinal flow patterns might be informative, and our findings are consistent, with the large body of literature that shows how instantaneous heading can be computed directly from retinal flow [20, 21, 23–25, 57, 70]. A key focus of the present study is the analysis of how this pattern varies through the gait cycle. Much of the work on optic flow either implicitly or explicitly assumes that optic flow is primarily relevant to steering to a distant goal, and so the experimental simuli tend to use use constant velocity straight line motion that is poor representation of the visual motion stimulus experienced during natural locomotion. Some work has suggested that heading estimates rely on information accumulated over time [38, 71, 72]. In particular, recent work by Burlingam and Heeger [39] demonstrates that instantaneous optic flow is insufficient for heading perception in the presence of rotation, and showed that heading judgments in the presence of rotation can be made on the basis of the time-varying evolution of the flow (that is, the retinal acceleration field). Although their stimuli did not mimic the kind of time variations generated by natural gait patterns, their finding that observers are sensitive to the temporal derivative of the retinal velocity field has an interesting relationship to the present discussion of divergence and curl. Divergence and curl are operators over $\nabla$ (referred to as "del"), which is defined as the 2 dimensional spatial gradient (derivative) of a vector field (in our case, the retinal velocity field). This means that our study examined ways to determine heading by way of the *spatial* derivative ($\nabla$) of retinal flow, where as Burlingham and Heeger solve a similar problem by way of the *temporal* derivative (acceleration) of the same vector field. Thus, despite the obviously substantial differences between the two approaches, both suggest that the solution to the determining heading from retinal flow lies in the consideration of the higher order structure of the instantaneous retinal flow field—in our case, by considering the spatial derivative of instantaneous retinal flow and in Burlingham and Heegers case by considering the temporal derivative of flow. Although it remains an empirical question as to whether and how either cue might be used in practice during natural locomotion, redundancy is the key to robustness, so it is possible that both factors contribute to the visual control of natural locomotion.

In experiments that use a stationary observer and simulate direction on a computer monitor, the strong sense of illusory motion (vection) and accurate estimates of simulated heading indicate that humans are highly sensitive to full field optic flow (e.g. [10, 73]). However, it does not necessarily mean that subjects use this information to control direction of the body when heading towards a distant goal. The complex, phasic pattern of acceleration shown here derives from the basic biomechanics of locomotion [50]. In the absence of direct measurements of flow during locomotion, the magnitude of the effect of gait has not been obvious. Thus it may have been incorrectly assumed that the overall structure of optic flow during locomotion would be dominated by the effects of forward motion. Such a forward-motion-dominated might be derived from temporal integration of eye-movement-corrected, head-centered optic flow, but given the large and rapid variation in the head velocity shown here it is unclear if simple temporal integration would be sufficient for accurate heading estimates.

## 4.1 The use of optic flow during natural locomotion

Many of the primary challenges of human walking operate on much shorter timescales than steering towards a distant goal. Striding bipedalism is inherently unstable—successful locomotion requires a walker to constantly regulate muscle forces to support the mass of the body above the ground [74, 75]. In complex, rough terrain, these supporting muscle forces critically depend on the careful placement of the feet onto stable areas of terrain that are capable of supporting the walker's body while redirecting the momentum of the center of mass towards future footholds [34, 63]. Failure to place the foot in a stable location or failure to properly support the body within a step may result in potentially catastrophic injury.

Given that walkers place their foot in a new location roughly twice per second, it is reasonable to assume that the underlying visual-motor control processes for foot placement and balance are fast, reliable, and efficient. It is also reasonable to suppose that subjects are able to learn the dynamic evolution of the flow patterns they generate during locomotion and so can use retinal flow patterns as a control signal. Indeed, it has been clearly demonstrated that optic flow exerts an important regulatory influence on the short-term components of gait. Expansion and contraction of optic flow patterns entrain the gait of subjects walking on treadmills, with the steps phase-locked to timing of expansion and contraction in the flow patterns [76–78]. Additional research has shown that optic flow can speed up adaptation to an asymmetric split-belt treadmill [56]. Other evidence indicates that the FoE had a strong influence on posture but not on heading [79]. [80] and [81] have also demonstrated regulation of stepping by optic flow, and [54] has shown that optic flow perturbations produce a short-latency response in the ankle musculature. O'Connor and colleagues have shown that optic flow perturbations have a stronger effect when they are in a biomechanically unstable direction during standing and walking [82], and that treadmill walkers will rapidly alter their walking speed to match altered optic flow before slowly adjusting back to their biomechanically preferred speed [83]. Taken together this body of research indicates a robust and sophisticated relationship between optic flow and the regulation of the short timescale biomechanical aspects of locomotion.

The data we present here do not directly demonstrate a role of retinal flow in the visual control of locomotion. However, all these studies suggest that the retinal optic flow cues described in this paper function as the control variables for stepping. In a study by Bossard et al [84], viewpoint oscillations consistent with head movements were added to a radial optic flow simulating forward self-motion. This was found to influence perception of distance travelled (as opposed to heading). This finding suggests that subjects are sensitive to the oscillatory effects of optic flow, and can integrate some aspects of flow information over the gait cycle and over longer time periods. However, this effect was only observed when the subjects were stationary, and not when they were walking on a treadmill [85]. This suggests that the context—whether walking or not—is an important modulating factor in the way that flow is used.

## 4.2 Optic flow, visual direction, and the control of locomotor heading

The act of steering towards a goal does not necessarily require the use of optic flow. [86] proposed that the perceived location (visual direction) of a target with respect to the body is used to guide locomotion, rendering optic flow unnecessary. Perhaps the strongest evidence for the role of optic flow in control of steering towards a goal is the demonstration by [87] who pitted visual direction against the focus of expansion in a virtual environment, where walkers generate the flow patterns typical of natural locomotion. They found that although visual direction was used to control walking paths when environments lacked visual structure (and thereby lacked a salient optic flow signal), optic flow had an increasing effect on paths as environments became more structured. The authors interpreted this result to mean that walkers use a

combination of visual direction and optic flow to steer to a goal when the visual environment contains sufficient visual structure. This is puzzling in the context of our findings, since the [87] experiment used a fully ambulatory virtual environment, so the head-centered optic flow experienced those subjects would have had the same instabilities described here. How then can we reconcile these results?

In the [87] experiment, the FoE and visual direction were put in conflict using a virtual prism manipulation that displaces the visual direction of the goal but not the FoE. However, in light of the current exploration of the structure of fixation-mediated retinal flow, it seems that this prism manipulation would also induce a change in the retinal curl subjects experienced during forward locomotion. To see why, imagine a walker (without prisms) moving forward while fixating a point that is directly on their travel path. As described above, this walker would experience no net curl at their point of fixation. Now imagine the same situation for a walker wearing prism glasses that shift their visual environment 10 degrees to the right—In that case, an individual walking forward while fixating a point that is directly on their travel path would experience retinal flow consistent with fixating a point that was 10 degrees to their right. Therefore, the walker would experience a retinal flow field with counter-clockwise rotation (negative curl) as in Fig 6C and S6 Video. If walkers use this retinal flow cue to control their foot placement, this effect might drive them to turn their body in the opposite direction within a step, in order to null this illusory retinal rotation/curl, resulting in a straighter path towards the goal. Put another way, retinal curl provides a moment-to-moment error signal that the walkers could use to actively correct for the effects of the prism. This hypothetical locomotor effect would be most pronounced in environments with salient optic flow, which could explain why the authors found that subjects walked in straighter lines in visually structured environments. In this interpretation, the walker's act of directing gaze towards their goal provides an egocentric heading cue to help them steer towards their target while retinal optic flow provides an error signal that may be used to control their locomotion and balance on a shorter timescale. Thus the present focus on the retinal flow and its evolution over time may help unify research on longer-timescale control of steering towards a goal [88] with the shorter-timescales associated with control of the mechanical aspects of balance and trajectory within a step.

## 4.3 Cortical involvement in the perception of optic flow

One of the insights from the observations in this study is that the stimulus input to the visual system is critically dependent on both the movement of the body and the gaze sampling strategies, especially in the case of motion stimuli (see also, e.g. [30]). Gaze patterns in turn depend on behavioral goals. In rough terrain gaze is mostly on footholds 2–3 steps ahead, whereas in regular terrain gaze is mostly directed to more distant locations [34]. This changes the pattern of flow on the retina as demonstrated. If humans learn the regularities in the flow patterns and use this information to modulate posture and control stepping, investigation of the neural analysis of optic flow ideally requires patterns of stimulation that faithfully reflect the patterns experienced during natural movement. A variety of regions in both monkey and human cortex respond to optic flow patterns and have been implicated in self motion perception, including MT, MST, VIP, CSv [13, 15, 16, 89–91] and activity in these areas appears to be linked to heading judgments. However, there is no clear consensus on the role of these regions in the control of locomotion. For example, in humans, early visual regions and MT respond about equally to motion patterns that are inconsistent with self-motion (for example, when the FoE is not at the fovea) as they do to consistent patterns. MST had intermediate properties [16]. Similarly in monkeys selectivity for ego-motion compatible stimuli was never total in a range of areas including MST and VIP [92]. [93] also conclude that MST is not critical for perception of self-

motion. In contrast, [12] conclude that area MST and PIVC/VPS contribute to heading perception, but not VIP. Since a variety of different motion patterns have been used in all these experiments, it may be necessary to simulate the consequences of self-motion with stimuli that are more closely matched to the natural patterns, and take the kinematics of the body movements into account, in order to reach more definitive conclusions. Many neurons within MST respond vigorously to spiral patterns of motion, which is notable in light of the ubiquitous spiral patterns that appear in the present dataset (e.g. S3 and S12 Videos).

Interestingly, although neurons in both MST and VIP signal heading, they do not strongly discriminate between the retinal patterns generated by simulated, as opposed to real, eye movements [90, 94]. This suggests that the information critical for heading is computed in a retinal reference frame [90, 95]. Similarly, [17] used a decoding model to estimate heading from neural activity in MST of monkeys looking at translational flowfields and noted that their reliability dropped every time the monkeys made a saccade The authors argue for a 'compression' mechanism to explain this effect, but in light of the present discussion, this result may be indicative of a neural system meant to extract heading on a per-fixation basis. This result is consistent with the suggestion of the present paper that the retinal patterns contain the critical information. The use of retinal flow for the control of movement requires that a walker has a notion of the body-relative location of their current fixation [60], which might explain the well-established connection between vestibular cues and optic flow parsing [44].

## 5 Conclusion

Gibson's critical insight on optic flow was that an agent's movement through their environment imparts a structure to the available visual information, and that that structure can be exploited for the control of action [5]. In the intervening years, a large and fruitful body of research built upon this insight. However, the way optic flow is used by the visual system is difficult to intuit without direct measurement of the flow patterns that humans generate during natural behavior. Advances in imaging technology and computational image analysis, together with eye and body tracking in natural environments, have made it easier to measure and quantify these complex aspects of the visual input. Examination of the retinal flow patterns in the context of fixating and locomoting subjects suggests a change in emphasis and reinterpretation of the perceptuomotor role of optic flow, emphasizing its role in balance, short term steering, and foot placement rather than in control of steering toward a distant goal.

While many methods exist to compute instantaneous heading from the retinal flow field, a consideration of these patterns relative to the gaze point through the gait cycle provides a different context for the way the retinal flow information is used to control real-world, natural locomotion.

## 6 Methods and materials

### 6.1 Ethics statement

The activities described in this manuscript were approved by the Institutional Review Board of the University of Texas at Austin (Study number: 2006–06-0085). Informed consent was obtained from each subject prior to data collection.

### 6.2 Experimental subjects

Three human subjects participated in this experiment (1 female, 2 male; mean age: 28.7 +/-5 years, mean height: 1.79 +/- .14 m, mean weight: 78.3 +/- 18.8 kg, mean leg length: .96 +/- .68 m). One of the subjects was the first author of this manuscript.

## 6.3 Equipment

Subjects' gaze was tracked using a Pupil Labs mobile eye tracker (Pupil Labs, Berlin, Germany). Each eye was recorded at 120Hz with 640x480 resolution (the eye cameras), while an outward facing camera mounted 3cm above the right eye (the world camera) recorded at 30Hz with 1920x1080 resolution and a 100degree diagonal field of view. Subjects' eyes were shaded using roll-up optometrist sunshades that covered the eyes and eye cameras but left the world camera uncovered. This method of shading the infrared eye cameras from sunlight was less robust than the full IR blocking face shield used in [34], but was necessary to prevent the computational video analysis algorithms (described below) from being affected by reflections on the inside of the mask. For this reason, data collection was conducted during a time of day when the walking path was mostly shaded from direct sunlight. Kinematics were recorded using the Motion Shadow full body motion capture system using inertial measurement units recording at 100Hz. Raw data were initially recorded on a backpack-mounted laptop worn by the subject and later post-processed using custom code written for Matlab (MathWorks, Natick, MA, USA).

## 6.4 Experimental task

Subjects walked along two separate paths in the greenbelt of Austin TX, USA– A flat, tree-lined path consisting of mulched woodchips in the Shoal Creek trail in Pease District Park (the "Woodchips path") and a rough, rocky creekbed consisting mostly of large boulders (the "Rocky path"). The Woodchips path was selected because it was flat enough that foot placement did not require visual guidance, but was visually textured enough for the optical flow detection algorithms to detect visual motion (as opposed to something like a concrete sidewalk which lacks sufficient visual texture for the optic flow algorithms to 'latch onto', resulting in empty patches in the resulting flow fields). The Rocky path was the same path used in the Rough Terrain condition of [34].

Subjects walked from start to the end of the Woodchips path while following one of three sets of instructions—Free walking, Ground looking, and Distant Fixation. In the Free Walking condition, subjects were instructed to simply walk from the start to the end location without any explicit instructions for what to do with their eyes. In the Ground Looking condition, subjects were asked to walk while looking at the ground at an approximately fixed distance ahead. In the Distant Fixation condition, subjects were asked to walk while maintaining fixation on a self-selected distant object that was roughly at eye height (switching targets as necessary to maintain distant fixation). This condition was intended to most closely match the psychophysical tasks involved in previous research on optic flow during locomotion, i.e. fixation on a distant object while moving forward without head movement. Locomotion in the Rocky trail was challenging enough that subjects were not asked to complete a secondary task. They were simply asked to walk from the start to end position at a comfortable walking pace.

Subjects completed three out-and-back walks in the Woodchips path, for a total of 6 trial/walks in that condition. There were two repetitions of each condition in the Woodchips (one per walking direction). Subjects completed 4 out-and-back walks on the Rocky path, for total of 8 trial/walks. Because the woodchips path was significantly longer than the rocky path, a similar amount of data was collected in each condition.

## VOR-based calibration and data post-processing

We used a procedure analogous to the VOR-based calibration method developed in [34], with some alternations due to the different in eye tracker. The Pupil Labs tracker used in this study estimate gaze for each eye using 3D spherical eye models generated within the coordinate

frame of each eye camera. Using the procedure described below, the gaze estimates for each eye were rotated to align with the reference frame of the full-body kinematic estimates from the IMU-based motion capture system.

The calibration procedure was completed at the start of data recording for each terrain condition for later processing. Subjects stood on a calibration mat (a black rubber outdoor carpet) that had marks for the subjects' feet, a high visibility marker located 1.4 m ahead of the vertical projection of the midpoint of the ankle joint, and a 0.5 m piece of tape at the same distance. Following the experimenter's instruction, subjects maintained gaze on that point while slowly moving their heads up/down, left/right, and along the diagonals. In addition to help determine the subject's 3D gaze vector by relating eye and head movements, data from this portion of the record were used to calibrate the eye tracker (similar to the "head tick" method described in [96], except that our subjects moved their heads smoothly).

The post-processing procedure to determine the subjects' world-centered 3D gaze vector is described below. Note that because each eye was calibrated independently, so any alignment between the gaze vectors of the right and left eye is an indication that the calibration was completed accurately. The final gaze vectors shown in laser skeleton videos (e.g. S1 and S13 Videos) are shown converging to a single point at the mean of each eye's gaze/ground intersection. The individual ground intersection of the right and left eye are visible as magenta and cyan points on the ground (respectively) of those videos. The agreement between these points is generally good and they tend to be more aligned when gaze is directed to the ground near to the subject (i.e. at the same distance the point used to calibration the eye trackers). It is unclear whether the disagreement between the two eye's gaze/ground intersection point is the result of actual differences in gaze direction of each eye or if it is the result of error in the eye tracker's gaze estimates.

Step 1 **Align timestamps of Pupil eye tracker and Shadow IMU data**. Because the eye tracker and the motion capture system were being recorded on the same backpack mounted laptop, the timestamps from the two systems were already synchronized because they were querying the same internal clock. Aligning the two data streams was a relatively simple matter of ensuring that both systems calculated time relative to some external temporal reference (rather than some unknown internal reference, such as "computer boot time").

Step 2 **Resample data from eye tracker and IMU to ensure constant 120Hz framerate**. Kinematic data from the Shadow system (recorded at 100Hz) were upsampled to 120Hz to match the framerate of the eye cameras. Following this, the left eye, right eye, and kinematic data streams were resampled to synchronize the three data streams.

Step 3 **Estimate eye center coordinates relative to head position estimates from Shadow system**. This location will serve as the center for the spherical eye model generated by the Pupil Labs eye tracker.

Step 4 **Situate 3D spherical eye model from Pupil tracker onto eye center estimate calculated in step 3**. Because this eye model was generated in the reference frame of the relevant eye camera, gaze vector orientations will be arbitrary when placed in the world-centered reference frame of the kinematic data from the Shadow system. The next step will align these gaze vectors with the subejcts' world-centered gaze.

Step 5 **Use data from VOR calibration task to rotate gaze data from Pupil tracker to align with the subjects' world-centered gaze using unconstrained optimization (MATLAB's fminunc function)**.

(a) Begin with a starting guess where the Euler angle rotation of the gaze data is [0 0 0].

(b) Rotate all gaze data by this rotation, and then rotate each gaze vector by the subject's head orientation on the frame that it was recorded.

(c) Calculate intersections between each gaze vector and the ground plane. If a gaze vector does not intersect the ground plane, truncate it at 10 m.

(d) Calculate error of this camera alignment rotation, defined as the mean distance between the intersection point of each gaze vector and the calibration point that subjects were fixating (located 1.4 m ahead of the vertical projection of the subject's ankle joints).

(e) Use fminunc to minimize this error by optimizing the Euler angle rotation to apply to the gaze vectors prior to applying the rotation specified by the subject's head orientation. The correct orientation will cause subjects' head rotations to cancel their eye movements to maintain the gaze vector alignment with the calibration point (that is, the correct gaze alignment rotation will preserve VOR-based eye compensation for head rotation).

Step 6 **Once the correct gaze alignment rotation has been identified, apply it to all subsequent gaze data from this recording**

Step 7 **Rotating each gaze point according to the subject's head orientation on that data frame**.

Step 8 To verify correct calibration, examine the intersection between the gaze vectors and the ground plane during the VOR calibration. If the head and eye data are properly calibrated, their rotations will cancel eachother out, resulting in a tightly clustered patch of gaze/ground intersection points around the calibration point

## 6.5 Video analysis and optic flow estimation

We used the Matlab Camera Calibration toolbox to estimate the lens intrinsics of the head-mounted world camera of the Pupil tracker (3 radial distortion coefficients with skew correction). This method utilizes a checkerboard of known size to determine the distortion caused by the wide angle lens of the camera. This calibration effectively allows us to treat the images from the head mounted camera as if they were recorded by a linear pinhole camera. We then estimated the visual motion on each recorded frame of the head-mounted camera using the Deep Flow optical flow estimation algorithm [47] in OpenCV [97]. DeepFlow is a dense optical flow estimation algorithm that provides a motion estimate at each pixel of each frame of the analyzed video. Despite its name, this method does not rely on deep learning methods to detect optic flow. Internal testing found that deep-learning based methods (e.g. FlowNet) are prone to biases likely due to their training data—Placing the head-centered videos in a large black swuare resulted in FlowNet reporting non-zero flow in the complete blank area (e.g. extrapolating outward from the actual motion in the video. Therefore, although these neural network based algorithms produce smooth and aethetically appealing flow estimates, they are not actually directly measuring visual motion in the scene and so are not appropriate for use as a scientific research method.

**6.5.1 Estimating retinal reference frame.** In order to estimate the visual stimulus in a retinal reference frame, each recorded frame from the undistorted world camera was projected onto a sphere and placed so that that subjects' point of regard on that frame was aligned with

0,0 in the retinal reference frame. To account for inaccuracy in this eye tracker, fixations were "idealized" by adjusting the placement of the image to null any residual motion detected at that point of fixation. (Inspection of the videos revealed only minimal residual motion during fixation, see Methods and materials.) In this way, the retinal reference frame videos provide an estimate of the visual motion that was incident to the subjects' retina, assuming "perfect" fixation and a spherical pinhole camera model of the eye.

## 6.6 Optic Flow simulation with a spherical pinhole camera model of the eye

We created a geometric simulation to provide a more nuanced picture of the way that the movement of the body shapes the visual motion experienced during natural locomotion. To estimate the flow experienced during various types of movements, a simulated eye model was generated using the following procedure. Most of the geometric calculations used in this model rely heavily on the Geom3D toolbox on Mathworks.com [98]

Step 1 **Define the groundplane as an infinite, flat plane with zero height**.

Step 2 **Define an evenly spaced grid of points on the ground plane**. These dots will eventually be projected onto the back of the simulated retina.

Step 3 **Define the spherical eyeball as a sphere with a radius of 12 mm** (the anatomical average radius of the human eye). In this model, one pole of the sphere is defined as the 'pupil' and the opposite pole is defined as the 'fovea'. Place the center of this eye model at the correct location in 3D space. This location is either determined from the recorded motion capture data (as in S1–S3 and S10–S13 Videos) or by determining a prescribed path for the eye to follow (as in S4–S9 Videos)

Step 4 **Rotate the eye model to face the fixation point on that frame**. Orient the eye so that there is a line passing through so that a line passing through fovea and pupil will also pass through the fixation point on the ground. This rotation was defined so as to minimize torsion about the optical axis. Our inability to estimate torsion is a notable weakness of the present methodology, especially given the prevalence of Curl in our analysis. We grudgingly neglect torsion for two related reasons—First, no modern mobile eye tracker reports ocular torsion, and second there is limited scientific information about ocular torsion during natural behavior. Our decision to hold torsion at zero is a standard simplification derived from Listing's law, but it is known that Listing's Law does not apply when VOR is active (as it is during virtually *all* natural behaviors) [99]. As the purview of modern visual neuroscience extends to cover natural behavior, we will need to develop better methods and theories to record and understand ocular torsion, e.g. [100].

Step 5 **Define the field of view of the eye by projecting a cone from the pupil determining the intersection points between this cone and the flat ground plane**. For this study, we defined the field of view cone to have a 60 degree radius.

Step 6 **Project all the groundpoints within the field of view through the pupil and onto the back of the 3D spherical retina**.

Step 7 **Resituate projected points onto a 2D polar plot** where the vertical axis (pi/2) defines the anatomical superior of the eye, and the eccentricity is defined as the "great circle" distance of the projected point from the 'fovea' of the spherical eye. Keep track of these projected locations across successive frames, to allow for calculation of optic flow on later steps.

Step 8 **Calculate flow per frame by determining the distance between the projection of each point on successive frames**. The flow on the first frame is defined to be zero everywhere. If a particular point was not within the eye's field of view on the previous frame, flow at that point is undefined on the first frame that point became visible.

Step 9 **Use the scatteredInterpolant function in Matlab to define an evenly spaced vector field across the retina that shares the structure of the projected-dot flow field determined in the previous step**. This step is necessary to be able to calculate the Curl and Divergence of the vector field.

Step 10 **Calculate divergence and curl of this evenly spaced retinal velocity grid using the divergence and curl functions in Matlab** (see **Curl and Divergence** section below).

Step 11 Spend several years fastidiously creating overcomplicated videos to highlight various features of the resulting flowfields (optional)

**6.6.1 Estimating and tracking the focus of expansion.** To estimate the location of the focus of expansion on each frame, each frame from world camera was first processed by the DeepFlow optical flow algorithm described above. This method provides a motion estimate for each pixel of the video frame, providing a 2 dimensional vector field with the same dimension as the original video for each recorded frame. To track the Focus of Expansion (FoE) in each frame, this vector field was first negated (all vectors were multiplied by -1), which effectively transforms the FoE from a repellor node (vectors pointing away from the FoE) into an attractor node (vectors pointing towards the FoE). Then, a grid of particles was set to drift on this negated flow field using the streamlines2 function in Matlab. The paths traced by these particles provide information about the underlying structure of the optic flow on each frame, represented as purple/white lines in Fig 2 and S3 and S12 Videos. These streamlines represent the line integrals of the optic flow vector field measured on each frame.

The location of the FoE was determined by keeping track of the final fate of each of the drifting particles and tagging the location where the majority ended up as the pixel location of the FoE on that frame. On frames where fewer than 50% of particles could be found in a particular location on the video screen, it was determined that the FoE was not in the field of view of the camera on that frame (this could be verified visually by noting that the streamlines on those frames do not converge to a single point). To calculate the velocity of the FoE on each frame, we calculated the 2D distance traveled by the FoE on successive frames, multiplied by that by the field of the view of the camera (to convert to Degrees Visual Angle) and then divided by the framerate (to convert to Degrees per second).

To ensure that this method was capable of detecting a stable FoE (that is, to ensure that the high velocity of the FoE that we record was not just an artifact of the detection method), we applied this analysis to the video from a DJI Phantom 4 quadcopter as it traveled along a straight horizontal and vertical flight path. The camera of this quadcopter is stabilized by a 3-axis gimbal mount, which is essentially a mechanical equivalent of the VOR-reflex. The resulting FoE is nicely stable, indicating that this method is capable of detecting a stable, low-velocity FoE in a video, if it is present (S14 Video).

## 6.7 The assumption of perfect gaze stabilization and the consequences of saccades

**The accuracy of gaze stabilization.** As shown by the videos and the eye movement records in Fig 1, subjects make saccades separated by periods where the eye is approximately

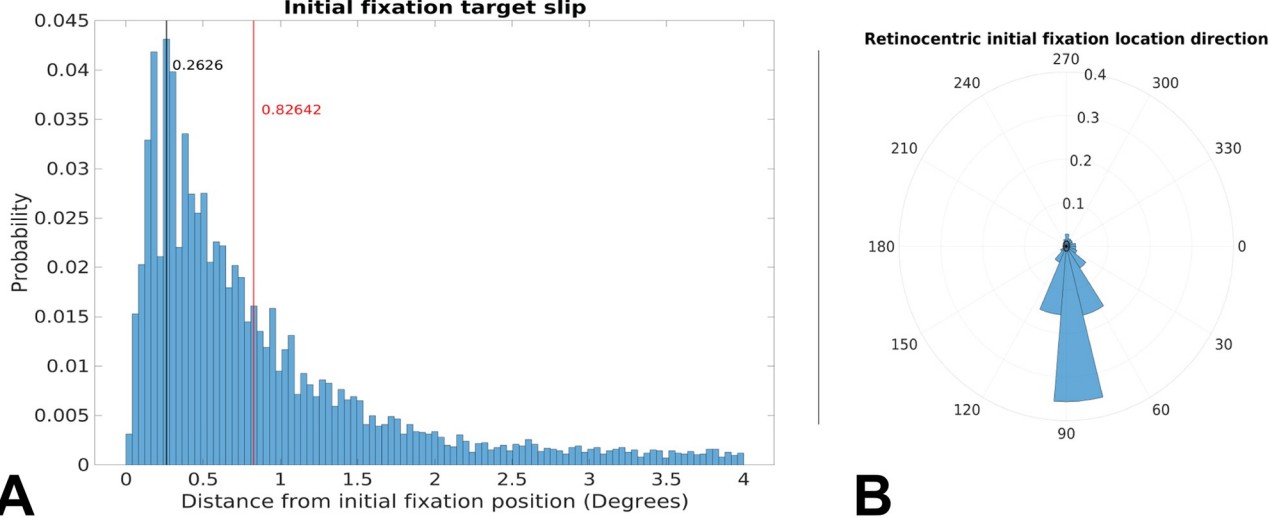

**Fig 8. Deviation from initial fixation location over the course of fixation.** Initial fixation location is computed and tracked over the course of each fixation, and compared to current fixation for duration of each fixation. Median deviation value is calculated for each fixation. The histogram captures the extent of variability of initially fixated locations relative to the fovea over the course of fixations, with most initially fixated locations never deviating more than 2 degrees of visual angle within the fixation. Long tail of distribution likely arises from erroneously labeled fixations where there are multiple small saccades. Fig 1B. Histogram of directions of retinal slip during a fixation. Most of the values are downward, as expected by a gain of less than 1.0.

stationary at a point in the world. Our calculations of retinal flow patterns are an idealized case. We assume a flat ground plane, which ignores the motion parallax information induced by irregularities in depth. We ignore eye tracker noise, and we also assume that gaze is perfectly stable during a fixation. If gaze is imperfectly stabilized, the retinal slip will introduce additional motion components and change the estimated div and curl as a function of time during the gait cycle. To estimate the consequence of imperfect stabilization, we used the DeepFlow algorithm on the video record to evaluate the actual image motion during a fixation. The histogram in Fig 8A shows a histogram of the slip magnitude during all the fixations. The mode of the distribution is approximately 0.25 deg of visual angle which indicates excellent stabilization. This was verified by frame-by-frame inspection of the video records. This histogram was cut off at 4 deg, which is the average saccade magnitude. Values greater than this would reflect no stabilization at all during a fixation. The tail of the distribution most likely reflect errors in the fixation finder which may have failed to differentiate fixations separated by small saccades. We also show that the direction of the slip is mostly downward, as expected from imperfect stabilization (See Fig 8B), but might also arise from a shifting in parallax errer as the fixated points draws nearer [96].

This analysis shows that stabilization was very good, and comparable to fixational stability observed under conditions where the head is stabilized. Consequently, it is probably not a major source of error in the calculation of retinal flow, especially given the inexact nature of foot placement. A foothold subtends about 2–4 deg laterally at a gaze angle of 45 deg to the horizontal (where much of the gaze is distributed) and visual information from anywhere within that region is most likely all that is required.

**Optical consequences of imperfect fixation (slip).** To estimate the consequences of additional retinal slip, we calculated how 1 deg of slip down and to the left during a 250 msec fixation would change the div and curl estimates. This would add 4 deg per sec of image motion at the fovea, which is larger than he median slip in the recorded data. This is shown in Fig 9, which revels modest changes to the flow patterns. It is hard to evaluate how significant that

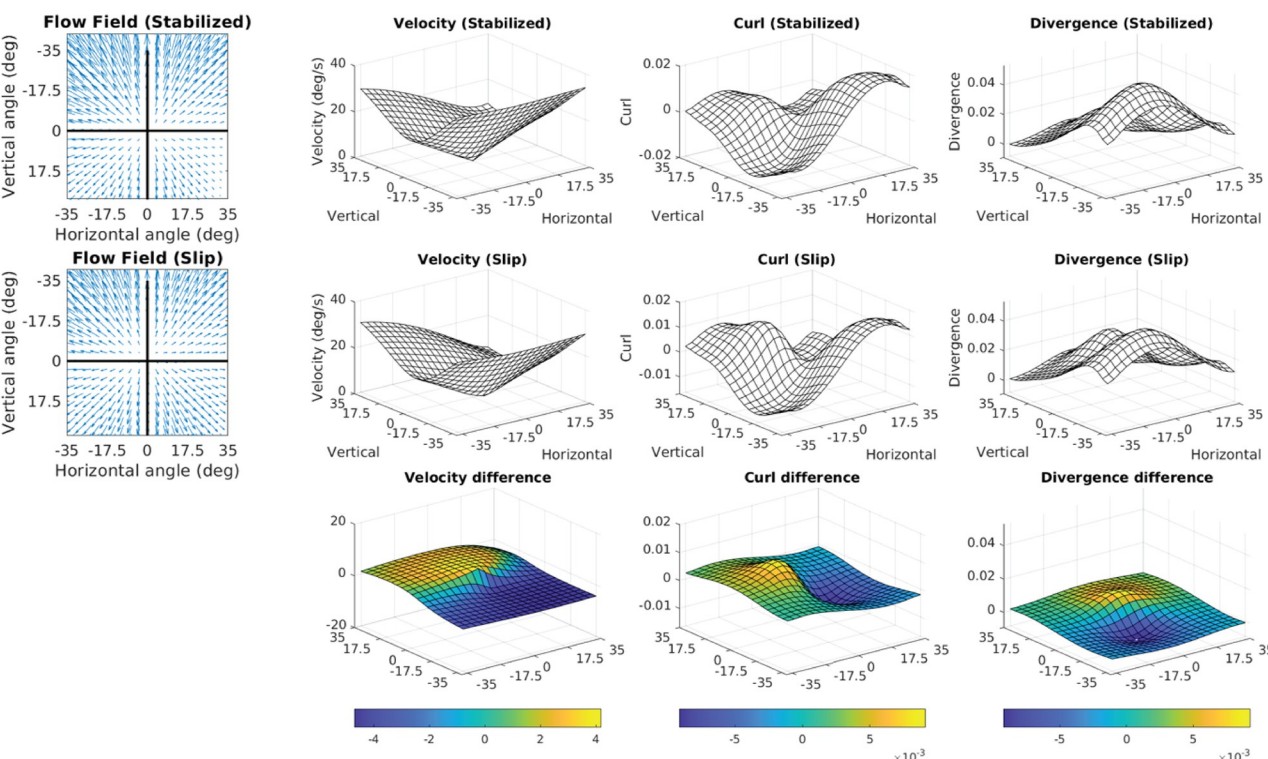

**Fig 9. A representative flow field during a perfect 250ms fixation (stabilized) and when undergoing 1 degree of slip.** An eye translation direction downwards and slightly to the left of fixation is examined. The top row shows the stabilized case (i.e. perfect fixation) and the second row shows the unstabilized case (i.e. retinal slip). The bottom row shows the difference between the two, calculated by simple subtraction. The two plots on the left show the flow fields for the stabilized and unstabilized cases, and are very similar qualitatively, with the focus of expansion slightly shifted in the opposite of motion slippage. The plots on the right are centered on the fovea, and show velocity fields, curl, and divergence values. During stabilization, velocities are low with 0 deg/s at the fovea due to simulated perfect fixation. For stabilization with slip, velocity at the fovea is 4.17 deg/s and velocities at other retinal locations are shifted either higher or lower depending on location, however the magnitude of this shift does not exceed the foveal slip. There are small changes in the curl and divergence fields but the patterns are very similar.

would be without knowing exactly how the curl and div signals are used which requires the development of a model and a set of assumptions. In this paper we simply measure the structure of the retinal motion patterns in the best-case scenario and suggest that it might be used as long as walkers can learn the regularities in the pattern. Showing that/how retinal flow information is used in practice is beyond the scope of the current paper.

**Visual motion generated by saccades.** In our analysis of retinal motion patterns, we have only considered the periods when the eye is approximately stable. These periods are interrupted by saccades, which generate a high velocity signal on the retina. The motion generated by saccades is not typically perceptually visible as a result of the mechanisms of saccadic suppression and visual stability, as is well known. The retinal image motion creates a blurred image and renders the information difficult to use for the control of action.

We have shown in other work that the location of gaze is tightly linked to the terrain complexity, and saccade timing linked to phase of the gait cycle. Therefore, we chose to examine the periods of stable gaze, where there is consensus that humans are collecting image motion necessary for controlling locomotion. The retinal image motion caused by saccades is of course of interest and must be dealt with by the visual system in ways that are not well understood, e.g. [55]. For completeness, we calculated the additional image motion on the ground plane engendered by a saccade (195 degrees per second, up and to the left) and this is shown in Fig 10.

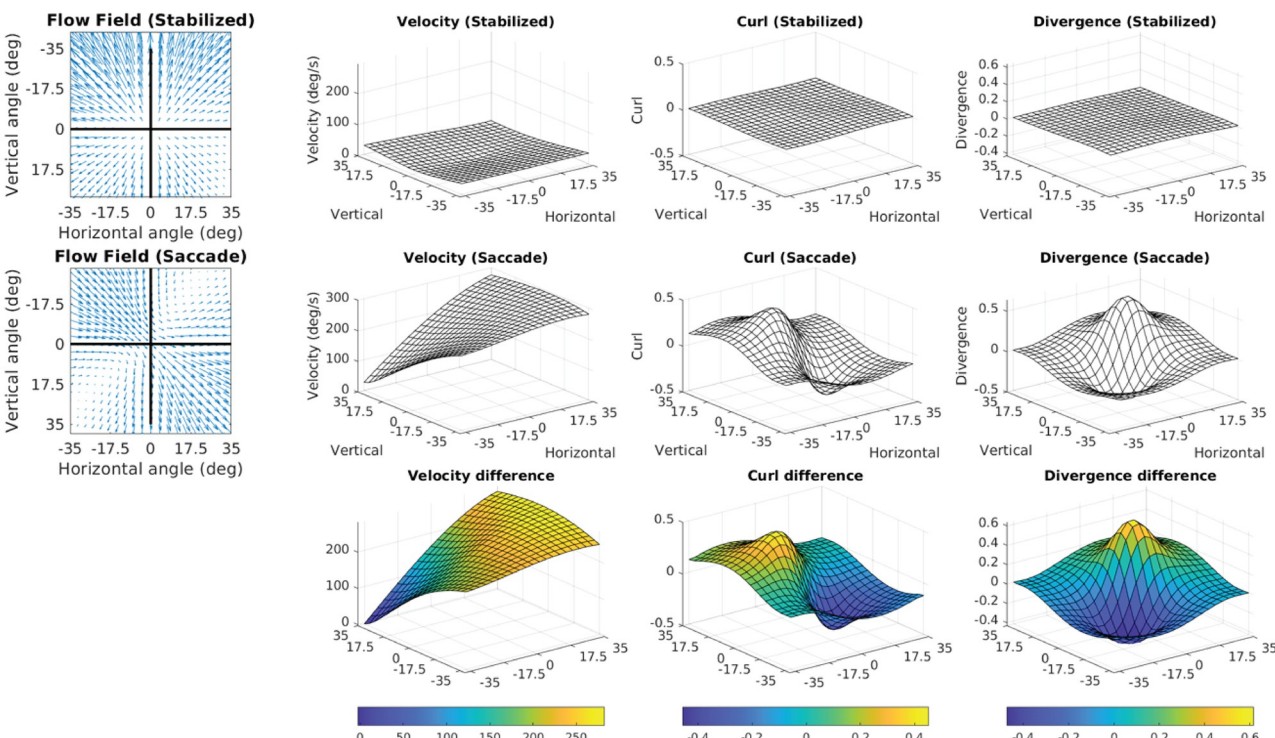

**Fig 10. Comparison of representative flow field during stabilization versus during a saccade (195 deg/s up and to the left).** The top row shows retinal motion while the eye is fixating and the second row shows motion during a saccade. The bottom row shows the difference for velocity, curl, and divergence plots between the motion during a fixation and motion during a saccade (that is, between the top and middle rows. The flow fields on the left show that when gaze is stable, motion is outwards, away from fixated location. During the saccade, flow is strongly influenced by the direction of the saccade (motion in the opposite direction is added). On the right side of the Figure, velocities are low with 0 deg/s at the fovea when gaze is stabilized. By contrast the saccade results in speed at the fovea equal to the speed of saccade (195 deg/s), with velocities at all other retinal locations increased by more than an order of magnitude. The saccade also results in increase in curl values by more than order of magnitude and alters the spatial structure of curl pattern. Similar to effect on curl values, divergence values change by order of magnitude, and spatial structure is altered. Note that the scale of the difference plot in the bottom row differs by a factor of over 10 in Fig 9.

## 6.8 Divergence and curl—Intuition and numerical calculation

We use divergence and curl operators to capture key factors of the retinal flow vector fields produced in Step Step 9: of Section 6.6 in Matlab using the following commands (Note, these commands also work in Octave, a free open-source software)-

```
[div] = divergence(retGridxx, retGridyy, retGridVelxx,
retGridVelyy]
[curlz, ~] = curl(retGridxx, retGridyy, retGridVelxx,
retGridVelyy]
```

. . .where retGridxx and retGridyy define a 2-dimensional grid of evenly spaced points in retinal coordinates and retGridVelxx and retGridVelyy encode the estimated 2-dimensional velocity of visual motion at the location of each grid point. div is the numerical divergence of the retinal flow field. curlz represents numerical curl of the retinal flow field in the z-direction, defined to be orthogonal to the x-y plane defined by retGridxx and retGridyy (Note that although curl is a vector in the case of a 3D vector field components in the X, Y, and Z, in the case 2D vector fields like those in this study, curl is only measured relative to the Z axis). Although a full introduction to the concepts of curl and divergence is beyond the present scope, an intuitive sense of this topic may provide helpful orientation of the results of this study. For an excellent introduction to the intuition and mathematical bases of curl and

divergence, we recommend this YoutTube video by 3Blue1Brown—https://www.youtube.com/watch?v=rB83DpBJQsE

**Divergence intuition.** The divergence of a 2d vector field is intuitively equivalent to expansion (positive divergence) or contraction (negative divergence). In the case of flowfield showing pure expansion (positive divergence), flow velocity is proportional to position on the grid—e.g. a step in the positive X direction results in an increase in X velocity (and similar for the Y direction).

A canonical example of expansive flow and uniform positive divergence is shown in Fig 11A, which was generated with the following Matlab/Octave commands (neglecting aesthetic alterations)-

```
%expansive flow
xmax = 6; %range of vector field
xmin = -xmax;
x = xmin:xmax;
xx = meshgrid(x); %create mesh grid containing X values of
grid dots
yy = xx'; %the Y values are the transpose of the X values
q1 = quiver(xx,yy,xx,yy); %show vector field
hold on
s1 = streamline(xx,yy,xx,yy,xx,yy); %show drifting massless
particles
```

Another way to think of it is to imagine the vector field as representing flowing water and then considering what would happen to a grid of massless particles floating under the influence of that flow for a period of time—If a given region of the flowfield would wind up with more particles than when it started, then we would say that region has negative divergence. If a given region winds up with fewer particles than it started out, then we would say it has positive divergence. Note that an area can have positive divergence even if it does not include a pattern of radial expansion (i.e. like the one found at the origin of Fig 11A), because areas away from that center point would still eject particles faster than they would accumulate them. The vector

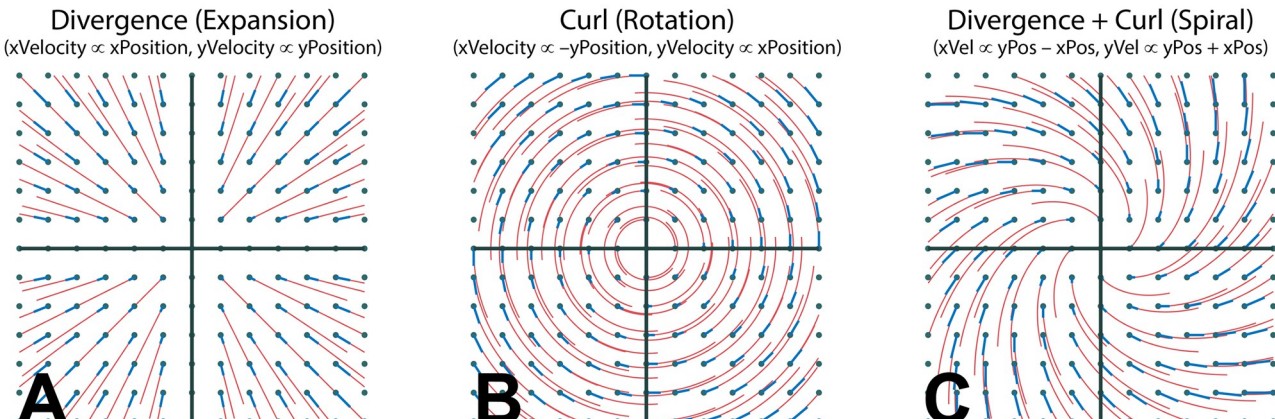

**Fig 11. Divergence, curl, and spiral flow patterns.** Canonical examples of Divergence (expansion), Curl (rotation), and the combination of divergence and curl (which produces spiral patterns. In the case of flowfield showing pure expansion (A—positive divergence), flow velocity is proportional to position on the grid—e.g. a step in the positive X direction results in an increase in X velocity (and similar for the Y direction). In the case of pure rotation (B—Curl) flow velocity increases in the orthogonal direction of the step—so a step in the positive X direction yields an increase velocity in the positive Y direction (for clockwise rotation/positive curl) or the negative Y direction (for counter clockwise rotation/negative curl). Summing the velocities of the divergence and curl flow fields results in spiral motion (C).

field defined in the code sample above has a uniform divergence value of 2 across the entire field.

**Curl intuition.** The curl of a 2-dimensional vector field is intuitively analogous to a measure of 'rotation' at each point in a vector field, with counter clockwise rotation denoting positive curl and negative rotation denoting negative curl. In the case of pure rotation (Curl) flow velocity increases in the orthogonal direction of the step—so a step in the positive X direction yields an increase velocity in the positive Y direction (for clockwise rotation/positive curl) or the negative Y direction (for counter clockwise rotation/negative curl).

A cannonical rotational flowfield shown in Fig 11B was generated with the following Matlab/Octave commands (carrying over variables defined in the Divergence example above) -

```
% rotational flow
%counter clockwise rotation (postive curl)
q1 = quiver(xx, yy, -yy, xx);
hold on
s1 = streamline(xx, yy, -yy, xx, xx, yy);

%clockwise rotation (negative curl)
q1 = quiver(xx, yy, yy, -xx);
hold on
s1 = streamline(xx, yy, yy, -xx, xx, yy);
```

Note that, similar to divergence, a non-zero curl/rotation value does not require a 'circular' rotation pattern (like that found at the origin on Fig 11B)—the flow defined in the code snipped above has a uniform Curl value of 2 (or -2) across the entire flow field. To see why, imagine the rotation of a pinwheel placed in a particular region of the vector field. No matter where you place the pinwheel in the vector field showin in Fig 11B, the pinwheel would always rotate in the same (counter clockwise) direction.

**Spiral flow.** The spiral flow patterns that are ubiquitous in the videos and animations shown in this paper arise from a combination of expansive androtational vector fields. The spiral flow pattern shown in Fig 11C was created by simply adding the vector velcocities shown Fig 11A and 11B -

```
% spiral flow
q1 = quiver(xx, yy, xx-yy, yy+xx);
hold on
s1 = streamline(xx, yy, xx-yy, yy+xx, xx, yy);
```

Note that this combination of the expansive and rotational vector fields into the spiral pattern shown in Fig 11C does not affect the divergence and curl values of the resulting flow field. The divergence and curl of the flow field created by the code snippet above is also 2 everywhere, just as it was in the pure expansion and pure rotation flow fields it comprises.

## Supporting information

### Video captions

This section contains information about this manuscript. It is a companion to the information listed in Table 1 of the main document, and provides additional information about the contents of each video.

The header for each video description follows the format of Table 1 -

```
Video ID/URL – Eye Trajectory/Video source – Data type – Short
Description
```

Where **Video ID/URL** denotes the ID for this video followed by a link to the video hosted on YouTube. **Eye Trajectory / Video source** denotes the source of the data represented in this video, e.g. if the data is derived from real-world recordings of subjects' gaze (measured by a Pupil Labs binocular eye tracker) and full-body kinematics (measured by a Motion Shadow IMU-based motion capture suit) or if it is based on a simulated eye trajectory. **Data Type** refers to the kind of data represented in a given video, e.g. a reconstructed 'laser skeleton' representation showing full body kinematics (skeleton) data with calibrated 3d binocular gaze direction (represented by lasers originating of the skeleton's estimated eyeball centers). A YouTube playlist of all the videos is available here—https://www.youtube.com/playlist?list=PLWxH2Ov17q5HRHVngfuMgMZn8qfOivMaf

An archival record of these videos is available on Figshare here—https://doi.org/10.25452/figshare.plus.17121686

**S1 Video.** https://youtu.be/3soholdtr7I- Real-world gaze/kinematics—Raw Laser Skeleton—Woodchips—Free Walking (Full speed, See Fig 1). A sample video of the data record. On the right is the view of the scene from the head camera (100 degree diagonal field of view) with gaze location indicated by the black crosshair. The black lines of the crosshair represent the gaze estimate for each recorded frame of the world camera (30Hz), while light grey crosshairs represent gaze estimates from the higher framerate eye cameras (120Hz) that occur between the world frames. The black circle has a radius of roughly 1 degree. Below the head centered video are the horizontal and vertical eye-in-head records, with blinks/tracker losses denoted by vertical gray bars. The high velocity regions (steep upwards slope) show the saccades to the next fixation point, and the lower velocity segments (shallow downwards slope) show the vestibular ocular reflex component that stabilizes gaze on a particular location in the scene as the subject moves towards it, resulting a characteristic saw-tooth appearance for the eye signal (without self-motion and VOR these saccades would exhibit a more square-wave like structure). On the left, the stick figure shows the skeleton figure reconstructed form the IMU-based full-body motion capture data. The skeleton data is integrated with the eye signal which is shown by the cyan and magenta lines. The representation of binocular gaze here shows the gaze vector from each eye converging on a single point (the mean of the intersection between the gaze vector from each eye and the groundplane). The original ground intersection of the right and left eye is shown as a magenta or cyan dot. The blue and red dots show the foot plants recorded by the motion capture system. The top left figure shows the scene image centered on the point of gaze reconstructed from the head camera as described in the Methods section. Note that the retina-centered video shows "idealized fixations," which assume zero residual image motion at the point of fixation (see Methods and materials).
(MP4)

**S2 Video.** https://youtu.be/jixjfxwKvQg- Real-world gaze/kinematics—Optic Flow—Vectors—Woodchips—Free Walking (Quarter Speed). The same video as S1 Video, but with visual motion represented by cyan arrows and played at 1/4 speed. The represented visual motion is derived from the Deepflow optical flow estimation algoritm (see Methods). The full data includes a motion vector estimate for each pixel of the recorded video, but this representation downsamples these flow vectors for easier viewing.
(MP4)

**S3 Video.** https://youtu.be/WAlzc8zgt4c—Real-world gaze/kinematics—Optic Flow—Streamlines—Woodchips—Free Walking (Quarter Speed). Same as S2 Video, but now with optic flow represented by streamlines representing the integral curves of the measured flowfield. Streamlines were calculated by inverting the flowfield on each frame and then allowing a grid of massless particle to drift across it. The white lines represent the path taken by each of

these drifting particles (see Methods). The yellow star represents the location of the Focus of Expansion (FoE) in the head centered reference frame (estimated as the location where at least 50% of the drifting particles wound up). The yellow vector extending from the head of the skeleton walker represents the 3D velocity vector of the estimated center of the walkers's cyclopean eye (scaled for visibility). The yellow line on the tip of the arrow represents the previous second of the head's velocity trace, with the yellow dots denoting measured frames from the world camera (30Hz, vs the higher framerate of the motion capture data, 100Hz). (MP4)

**S4 Video.** https://youtu.be/20kj4ew02aU—Simulated Eye Trajectory—Simulated Retinal Flow—Fixation Aligned with Path. Simulated retinal for for a spherical pinhole model of the eye that is moving forward at a constant velocity while maintaining fixation on a point that is aligned with its velocity vector. The central circle shows the view point of the eyeball, with a grid of ground dots projected onto the back of the eye through a pinhole pupil. Each dot is shown with a vector denoting its position relative to its position on the previous frame. Purple lines represent optic flow streamlines, calculated as in S3 Video. The representation to the left of this central circle shows the curl of the retinal flowfield calculated using Matlab's 'curl' operator (see Methods). The thick black contour line represents points with zero curl, while the green line represents the curl value at the fovea (the green foveal isoline is not visible in this video, as foveal curl equals zero when gaze is aligned with the velocity vector). Positive curl values (counter-clockwise rotation) are shown in blue, while negative (clockwise) rotaion is shown in red. Below and to the left this represenation, the curl values are represented spatially, with postive/negative curl shown as higher/lower vertical values. To the right of that is a histogram showing the probability of each curl value for each frame of the animation. The image below those plots shows the curl map for the retinal flowfield projected back onto the groundplane. The data representations to the right of the central circle are similar to those to the right, except that they represent the divergence of the retinal flowfield. (MP4)

**S5 Video.** https://youtu.be/HxkGI4MkQkw—Simulated Eye Trajectory—Simulated Retinal Flow—Fixation to Left of Path. Similar to S4 Video, but the simulated eyeball is fixating a point to the left of their current trajectory (i.e. it will pass to the right of the fixated point), resulting in counter clockwise retinal flow. (MP4)

**S6 Video.** https://youtu.be/CkDbErx-o-o—Simulated Eye Trajectory—Simulated Retinal Flow—Fixation to Right of Path. Similar to S4 Video, but the simulated eyeball is fixating a point to the right of their current trajectory (i.e. it will pass to the left of the fixated point), resulting in clockwise retinal flow. (MP4)

**S7 Video.** https://youtu.be/2X-vSIdEd3o—Simulated Eye Trajectory—Simulated Retinal Flow—Vertical Sin Wave. Similar to S4 Video, but the simulated eyeball is following a vertical sin wave trajectory. (MP4)

**S8 Video.** https://youtu.be/zjK1pQoks1w—Simulated Eye Trajectory—Simulated Retinal Flow—Horizontal Sin Wave. Similar to S4 Video, but the simulated eyeball is following a horizontal sin wave trajectory. (MP4)

**S9 Video.** https://youtu.be/nKmZA0T7-wc- Simulated Eye Trajectory—Simulated Retinal Flow—Forward Corkscrew. Similar to S4 Video, but the simulated eyeball is following a circular "corkscrew" trajectory (e.g. a combination of the trajectories in S7 Video and S8 Video).
(MP4)

**S10 Video.** https://youtu.be/SwErSYocvBs—Real-world gaze/kinematics—Simulated Retinal Flow—Woodchips—Ground Looking (Quarter speed). Similar to S4 Video, except now the eye trajectory is defined by motion capture data from the "Ground Looking" condition on the Woodchips terrain. Note that this representation shows "idealized" fixations, which assume zero velocity of the gaze vector's intersection with the groundplane. This idealization was introduced to account for erroneous motion imparted by drift from the IMU motion capture and error in the eye tracker.
(MP4)

**S11 Video.** https://youtu.be/jv8hS3MJoDQ- Real-world gaze/kinematics—Raw Laser Skeleton—Rocky Terrain (Full speed). Same as S1 Video, except this shows data from locomotion over rocky terrain (e.g. the 'Rocks' condition). Note that this is the same terrain that was used in the "Rough Terrain" condition of Matthis Yates and Hayhoe (2018).
(MP4)

**S12 Video.** https://youtu.be/36DwddZ8EKM- Real-world gaze/kinematics—Optic Flow—Streamlines—Rocky Terrain. Same as S1 Video, but showing data from the Rocks condition.
(MP4)

**S13 Video.** https://youtu.be/jwXwpRIWhwc- Real-world gaze/kinematics—Simulated Retinal Flow—Rocky Terrain. Same as S10 Video, but showing data from the Rocks condition.
(MP4)

**S14 Video.** https://youtu.be/WX8mgOoww9w—Quadcopter Gimbal Video—Optic Flow—Streamlines—Drone video validation. This video shows the optic flow streamline analysis on a gimbal stabilized video from a DJI Phantom 4 camera quadcopter. Note that the FoE of this video is relatively stable, indicating that this methodology is capable of detecting a stable FoE from a video taken by a moving camera.
(MP4)

**S1 Fig. Head rotational velocity.** Head rotational velocity throughout the course of the gait cycle. Measured similarly to the the acceleration values reported in Fig 4, but using rotational velocity measured by the triaxial gyroscope in the head-mounted IMU. Note that gyroscopes are generally noisier and less reliable than linear accelerometers.
(TIF)

## Acknowledgments

Special thanks to John Cormack for his advice regarding the computation analysis of fluid flow, which led directly to the development of the streamline method for tracking the focus of expansion.

## Author Contributions

**Conceptualization:** Jonathan Samir Matthis, Karl S. Muller, Kathryn L. Bonnen, Mary M. Hayhoe.

**Data curation:** Jonathan Samir Matthis, Karl S. Muller.

**Formal analysis:** Jonathan Samir Matthis, Karl S. Muller.

**Funding acquisition:** Jonathan Samir Matthis, Mary M. Hayhoe.

**Investigation:** Jonathan Samir Matthis, Karl S. Muller, Kathryn L. Bonnen.

**Methodology:** Jonathan Samir Matthis, Karl S. Muller, Kathryn L. Bonnen, Mary M. Hayhoe.

**Project administration:** Jonathan Samir Matthis, Mary M. Hayhoe.

**Resources:** Jonathan Samir Matthis, Mary M. Hayhoe.

**Software:** Jonathan Samir Matthis, Karl S. Muller, Kathryn L. Bonnen.

**Supervision:** Jonathan Samir Matthis, Mary M. Hayhoe.

**Validation:** Jonathan Samir Matthis, Karl S. Muller, Kathryn L. Bonnen, Mary M. Hayhoe.

**Visualization:** Jonathan Samir Matthis, Karl S. Muller, Kathryn L. Bonnen.

**Writing – original draft:** Jonathan Samir Matthis, Mary M. Hayhoe.

**Writing – review & editing:** Jonathan Samir Matthis, Karl S. Muller, Kathryn L. Bonnen, Mary M. Hayhoe.

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
