## [Decision Letter · Decision Letter 0]

24 Mar 2021

Dear Dr Matthis,

Thank you very much for submitting your manuscript "Retinal optic flow during natural locomotion" for consideration at PLOS Computational Biology.

As with all papers reviewed by the journal, your manuscript was reviewed by members of the editorial board and by several independent reviewers. In light of the reviews (below this email), we would like to invite the resubmission of a significantly-revised version that takes into account the reviewers' comments.

You will see that the reviewers felt that your work is very interesting. The authors should carefully consider the reviewers comments. In addition, I would like the authors to consider adding details on the spatial accuracy (in the environment tested) of the Pupil Lab eye tracker and Motion Shadow equipment in the Methods section, as both of these devices can be finicky, and the analyses rely on accurate measurements.

We cannot make any decision about publication until we have seen the revised manuscript and your response to the reviewers' comments. Your revised manuscript is also likely to be sent to reviewers for further evaluation.

Sincerely,

Daniel S Marigold

Guest Editor

PLOS Computational Biology

Thomas Serre

Deputy Editor

PLOS Computational Biology

Reviewer's Responses to Questions

**Comments to the Authors:**

Reviewer #1: This paper reports the results of a study in which participants' gaze were monitored while walking over terrain with a head mounted camera viewing the scene, allowing for the first ever recording of time-varying images on the retina during natural locomotion. The authors analyze the optic flow patterns, revealing interesting and highly non-trivial patterns that could be used as cues in guiding navigation.

This is a landmark study, sure to cause many people in the field to rethink what optic flow means, and how it is used during natural vision. A rich set of findings, beautifully analyzed, and the videos are fantastic. This is sure to get a lot of attention and motivate follow up studies, etc. It is a paper that really forces you to think as the results are not obvious.

Some general comments:

You describe multiple conditions for the locomotion, mulch vs. rocky terrain, how the subject is looking ahead, etc. But I don't see that the results of these different conditions are separated out, they all seem lumped together somehow. There is a figure inset in Figure 2, but I didn't see anything else. Is it because there are no appreciable differences?

Figure 2 shows a histogram of the rapid speeds at which the FOE moves in head centered reference frame. 1000 deg/sec seems remarkably high, perhaps worth commenting on this. Also it would be nice to see the corresponding histogram in the retinal case - it should be much sharper and peaked near zero right?

The manner in which images are cast into retinal coordinates seems a bit strange - i.e., assuming "perfect" fixation etc. I think you should define for the reader what perfect fixation means. I'm not sure just from reading everything in the methods that I could exactly replicate what the authors did, I think it could use spelling out more.

Section 2.1.3: "When gaze is properly stabilized on a location in the world during locomotion, the result will always be a pattern of radial expansion combined with rotation, centered on the point of fixation," - but in the intro I think you say this: "The resulting flow field will comprise both a rotational and a translational component," -- This doesn't seem consistent.

curl and divergence: I think it would be worth providing a short recap for those of us rusty on our calculus about what these things measure exactly, and how they are computed. Also I had thought curl produces a vector field, but in Figure 5 you show just a scalar field. And then there is the idea of "net curl" because the +'s cancel the -'s. Yet the divergence measure is just calculated locally and we don't have a "net div." Not sure how to think about all this, again a bit more mathematical explanation would be helpful here.

Section 2.2.3: "The point of maximum divergence encodes the body's over-ground momentum vector in retinotopic coordinates" - if this is true, which I suppose it is, then shouldn't you be able to show this from your data? i.e., you have the body speed and you know where they are looking, so you should be able to make a joint scatter plot of these things showing a linear relationship, no? I think it would be quite compelling and help to bolster the point.

Similarly for the curl patterns - you argue that this could be a cue, but the evidence shown is rather anecdotal, we just see a few select plots. How about just show a joint scatter plot of curl vs. where the subject is fixating?

Typos:

nulls image at the fovea,  nulls image motion at the fovea,

However, not matter  However, no matter

the the point  the point

line:

14 - this claim may be true but I think it needs a citation.

47 - 'judgements of their direction'

51 - 'no matter'

72 - 'apostrophy+t' misplaced

107 - period and semicolon together

185 - the left apostrophe is the right-side type

Figure 4 - should add labels of gamma, rho coordinates to figures, especially part a to demonstrate

276 - space before 'Because'

299, 313, ,377, 378- maybe bold or italic instead of **

364 - space before 'Some'

523 - 'Estimate'

548 - remove space before very last period

584,585 - consistent capitalization for woodchips and rocky

Reviewer #2: This paper is a major piece of work, extending classical descriptions of optic flow (vehicular motion) to its actual description during natural locomotion.

In order to make the contrbution "sharper", some concerns must be taken into account

1. As concerns the general paper's organization, discussion is rather large, compared to the paper itself. Parts of it might be inserted in introduction? For example, definitions of curl and divergence, that are central to the paper, were introduced by Koenderink. In the paper curl is introduced without being defined.

2. This is not an experimental paper per se (no real independent variable per se, besides maybe the terrain). This looks more as a high technicity instrumented observation and should be presented as such.

3. The authors started the discussion with "we have demonstrated....". This is not a demonstration but a description that leads to the suggestion that retinal flow and particularly curl motion play a critical role in heading control. This aspect of the paper is ground breaking. However, controlled experiments are needed to demonstrate this suggestion.

4. On a more functional level, the point is repeatedly made that retinal flow is the key input. What about head centered flow. In particular, is OKN totally irrelevant?

5. This point is somehow mentioned in conclusion, but how might some rythmical aspects of the visual consequences of walking be present (and used) in the flow (including the retinal one)?

6. page 5. "modal velocity across conditions of about 255 deg". Saccadic velocity? Just head velocity?

7. What is the necessity of the description of head velocity (figure 3)?

8. Page 11. The max retinal divergence is linked to the body's world centered velocity. But do we really use absolute values? Don't we real of relative measures (ttc etc.)?

Minor point:

A few typos across the manuscript

Reviewer #3: Matthis and colleagues have recorded (head-centered) visual optic flow during real self-motion. The concurrent measurement of eye movements allowed them to construct (offline) an eye-centered representation of optic flow during real self-motion. The results are (at least from my point of view) unexpected and really exciting. The authors argue that not the head-centered flow is the most appropriate for navigational purposes, but rather the eye-centered version of it. Nevertheless, in their conclusion, the authors question the role of optic flow for the sensory guidance of everyday locomotion.

This study has been a tour-de-force. Experimentally, the authors recorded eye, head- and body movements during self-motion in the real world. Given the different recording devices, synchronization of time series data is a first major challenge. After having solved that, the authors constructed a series of retinal images (videos) based on sequences of head-centered optic flow, the eye movement recordings and a spherical pinhole model of the eye. From my point of view, it definitely was worth the effort. Results are surprising and definitely have the potential to trigger a full new series of experimental and theoretical studies on visual self-motion processing. Having said this, I also must say that I see a number of points that need to be addressed.

Major

First and foremost, the authors have constructed the eye centered images based on the assumption that the gain of tracking movements of an earth fixed target is 1.0. Yet, as shown by Lappe and colleagues (J Neurophysiol, 1998), this gain typically is more in the order of 0.5. This difference has the potential to change the results a lot. I assume that eye centered optic flow fields are no longer as stable over time as suggested by the authors. This is critical and needs to be discussed.

Second and related: the authors have a long-standing experience in measuring eye movements in the real world. Hence, they know that humans make 2-3 eye movements per second. The rather stable eye centered optic flow - as documented here - is found during smooth eye movements, but not during saccades. This, again, is critical, especially for the discussion of the results. The authors must discuss the consequences of saccades on the structure of retinal flow fields and their implication for the use of optic flow for navigation.

Third: tracking of an earth fixed target during self-motion is not a “fixation”, but rather a smooth tracking movement. Hence, it could be called a smooth pursuit. This term, fixation, is used over and over in the manuscript. Yet, it is misleading. From my point of view, this does not call the overall results into question. But it has the potential to bring the discussion about the neural basis into a new direction. There are other brain regions involved in the control of smooth pursuit than of fixation. Hence, I recommend to reconsider the wording and adjust the discussion.

To be honest, I am not really happy with the discussion of the recent paper by Heeger’s group (PNAS, 2020). To my understanding, the authors of the current study argue that their div-operation, i.e. the spatial derivative, is somewhat similar to the temporal derivative (i.e. acceleration rather than speed) as found in Burlingham and Heeger, 2020. Physically (and mathematically), this is not the case.

The discussion of potential neural correlates of the properties as found here could be improved. As an example, the authors argue that – based on their results and results from Strong et al., 2017 – area MST is not critical for the perception of self-motion. This would also be in line with results e.g. from Wall and Smith (2008), who showed with fMRI in humans that hVIP but not hMST responds solely to visual optic flow which is compatible with self-motion. On the other hand, the group of Angelaki and DeAngelis argue that monkey area MST (and PIVC and VPS) but not area VIP is behaviorally relevant for self-motion perception (2016). I suggest that the authors elaborate a bit on this controversy. Not lengthy, but to make clear that the involvement of both areas is currently not clear and, hence, is heavily investigated. Furthermore, I suggest to consider neurophysiological studies by the groups of Lappe and Bremmer, who had tested responses in monkey areas MST and VIP in very similar contexts, i.e. forward self-motion and concurrent (real or simulated) eye movements (2010 and 2014).

Minor

There were quite a few typos and also references to supplementary material that does not exist. Examples are the legend of Figure 3 (page 7/31) and line 168. I recommend careful proof-reading of the revised version.

The manuscript could be shortened a bit. An example is the introduction (elaboration of the phylogenetically old VOR). In the same vein, the last, half-page paragraph of the introduction could be more or less skipped or shifted towards the discussion.

There were in-depth discussions in the Results section, which – to my taste – should be moved into the discussion. Examples are:

- Section 2.2.2, around line 243

- Section 2.2.3, lines 271 – 285

- Section 2.2.3, lines 292 – 316

**Have all data underlying the figures and results presented in the manuscript been provided?**

Reviewer #1: Yes

Reviewer #2: None

Reviewer #3: None

PLOS authors have the option to publish the peer review history of their article (what does this mean?). If published, this will include your full peer review and any attached files.

Reviewer #1: **Yes: **Bruno A. Olshausen

Reviewer #2: **Yes: **Daniel R Mestre

Reviewer #3: No
---

## [Decision Letter · Decision Letter 1]

14 Oct 2021

Dear Dr Matthis,

We are pleased to inform you that your manuscript 'Retinal optic flow during natural locomotion' has been provisionally accepted for publication in PLOS Computational Biology.

Best regards,

Daniel S Marigold

Guest Editor

PLOS Computational Biology

Thomas Serre

Deputy Editor

PLOS Computational Biology

As the authors can see, the reviewers were satisfied with the revisions. I have also gone over the manuscript and response letter and agree with their assessment. Unfortunately, we were unable to obtain the assessment from one of the original reviewers, which caused a delay in making a decision. After repeated attempts, I made the decision to proceed without that reviewer's assessment to not further delay a final decision.

Reviewer's Responses to Questions

**Comments to the Authors:**

Reviewer #2: None

Reviewer #3: The authors have addressed all my concerns adequately. I have no further comments.

**Have the authors made all data and (if applicable) computational code underlying the findings in their manuscript fully available?**

Reviewer #2: None

Reviewer #3: None

PLOS authors have the option to publish the peer review history of their article (what does this mean?). If published, this will include your full peer review and any attached files.

Reviewer #2: No

Reviewer #3: No

---

## [Editor Report · Acceptance letter]

4 Feb 2022

PCOMPBIOL-D-21-00167R1 

Retinal optic flow during natural locomotion

Dear Dr Matthis,

I am pleased to inform you that your manuscript has been formally accepted for publication in PLOS Computational Biology. Your manuscript is now with our production department and you will be notified of the publication date in due course.

With kind regards,

Olena Szabo
